# Long-term depression in neurons involves temporal and ultra-structural dynamics of phosphatidylinositol-4,5-bisphosphate relying on PIP5K, PTEN and PLC

Sarah A. Hofbrucker-MacKenzie[1], Eric Seemann [1], Martin Westermann[2], Britta Qualmann [1✉] &
Michael M. Kessels [1✉]

Synaptic plasticity involves proper establishment and rearrangement of structural and functional microdomains. Yet, visualization of the underlying lipid cues proved challenging. Applying a combination of rapid cryofixation, membrane freeze-fracturing, immunogold labeling and electron microscopy, we visualize and quantitatively determine the changes and the distribution of phosphatidylinositol-4,5-bisphosphate ($PIP_2$) in the plasma membrane of dendritic spines and subareas thereof at ultra-high resolution. These efforts unravel distinct phases of $PIP_2$ signals during induction of long-term depression (LTD). During the first minutes $PIP_2$ rapidly increases in a PIP5K-dependent manner forming nanoclusters. PTEN contributes to a second phase of $PIP_2$ accumulation. The transiently increased $PIP_2$ signals are restricted to upper and middle spine heads. Finally, PLC-dependent $PIP_2$ degradation provides timely termination of $PIP_2$ cues during LTD induction. Together, this work unravels the spatial and temporal cues set by $PIP_2$ during different phases after LTD induction and dissects the molecular mechanisms underlying the observed $PIP_2$ dynamics.

[1] Institute of Biochemistry I, Jena University Hospital — Friedrich Schiller University Jena, 07743 Jena, Germany. [2] Center for Electron Microscopy, Jena University Hospital — Friedrich Schiller University Jena, 07743 Jena, Germany. ✉email: Britta.Qualmann@med.uni-jena.de; Michael.Kessels@med.uni-jena.de

The majority of excitatory postsynapses in the central nervous system is located in small dendritic protrusions, termed dendritic spines. Synaptic plasticity processes involving dynamic changes in structure and organization of dendritic spines underlie the modulation of excitatory synaptic strength and are thought to be the basis for learning and memory[1,2]. Prominent among the modes of synaptic plasticity is a process called long-term depression (LTD), which reduces sensitivity and response to a given synaptic signal[3]. At the molecular level, LTD involves the removal of α-amino-3-hydroxy-5-methyl-4-isoxazolepropionic acid (AMPA)-type glutamate receptors from the postsynaptic density scaffold by endocytosis or lateral diffusion as well as structural rearrangements of the actin cytoskeleton resulting in a reduced spine volume[3,4].

During the last decades, great progress has been made in identifying and characterizing the protein machineries that control and mediate dendritic spine plasticity. Yet, knowledge about the involvement of membrane lipids—although they are major membrane constituents—is still sparse and often controversial or even contradictory[5,6]. Phosphoinositides are known for interacting with and regulating distinct membrane proteins and having a variety of cellular functions[7]. Phosphatidylinositol-4-phosphate (PI4P) and phosphatidylinositol-4,5-bisphosphate (here for simplicity referred to as PIP$_2$) are the most abundant phosphoinositides in cells. PIP$_2$ is enriched at the plasma membrane, where it constitutes less than one percent of the phospholipids[8]. In presynapses, crucial roles for phosphoinositides have been well-stablished and mainly comprise membrane trafficking functions[9–12]. In contrast, the physiological importance and specific role of PIP$_2$ in postsynapses are far less understood and controversial.

The concentrations of individual phosphoinositides are controlled by a complex set of specific lipid phosphatases, kinases and lipases that are themselves targets of a variety of signaling pathways[13,14]. Studies investigating a putative role of PIP$_2$ in especially LTD largely relied on manipulating these metabolic enzymes and yielded conflicting results[15–20]. The reasons for the (apparent) discrepancies may to a large extend include the inability to fix and to directly visualize PIP$_2$ at postsynaptic spines without altering its distribution and its undisturbed availability as lipid cue.

Synapses have particular functional and structural demands for compartmentalization, as established by distinct membrane nanodomains. Additionally, these synaptic membrane nanodomains require the ability to plastically adapt during postsynaptic rearrangements. To follow the hypothesis that these rearrangements during LTD might be brought about by temporal and spatial cues set by the important signaling molecule PIP$_2$, we applied rapid cryofixation, membrane freeze-fracturing, immunogold labeling and transmission electron microscopy (TEM) to visualize and quantitatively assess the distribution of PIP$_2$ in the plasma membrane of dendritic spines and in different subareas thereof at ultrahigh resolution. We thereby unveil the spatial and temporal cues set by PIP$_2$ during different phases after LTD induction and dissect the molecular mechanisms underlying the observed PIP$_2$ dynamics.

## Results

Previous studies attempting to unravel involvements of PIP$_2$ signaling cues in dendritic spine membranes during LTD yielded contradictory results[16,19]. As apparent discrepancies may have been caused by a lack of preservation of PIP$_2$ distribution, by artefactual PIP$_2$ quenching, by insufficient resolution and/or by other limitations, we aimed to set up an anti-PIP$_2$ immunolabeling of rapidly cryo-preserved freeze-fractured membranes analyzed by TEM. Liposomes with either PIP$_2$, phosphatidylserine (PS),

PI(3,4,5)P$_3$ (PIP$_3$), or PI(3,4)P$_2$ added were quick-frozen in liquid nitrogen-cooled propane/ethane, freeze-fractured and incubated with anti-PIP$_2$ and gold-coupled secondary antibodies to address whether it may be possible to preserve and detect the signal lipid PIP$_2$ in a membrane environment. Indeed, electron microscopical examinations of liposomes with added PIP$_2$ showed frequent immunogold labeling (29.32 gold particles/µm$^2$), whereas control liposomes did not (Fig. 1a–d). Quantitative evaluations showed that liposomes with PIP$_2$ added showed a more than 12-fold higher immunogold labeling density when compared to control liposomes, which merely offered a surplus of unrelated, negatively charged lipid head groups in form of PS addition (Fig. 1e; Supplementary Fig 1). These results clearly demonstrated that it is in principle possible to detect cryo-preserved PIP$_2$ in an intact lipid surrounding.

Further experiments showed that neither PIP$_3$ nor PI(3,4)P$_2$ was recognized by the anti-PIP$_2$ antibodies (Fig. 1c, d). The labeling densities observed at both of these additional controls were very low and did not exceed that of PS (Fig. 1e; Supplementary Fig. 1). Thus, neither inositides in general nor a phosphoinositide containing all potential antigenic features also present in PIP$_2$ but offering an additional phosphate (PIP$_3$) or a phosphoinositide displaying all features of PIP$_2$ but presenting one phosphate group in a different position (PI(3,4)P$_2$) were recognized by the anti-PIP$_2$ antibodies. The established immunodetection thus clearly was specific for PIP$_2$.

Quantitative determinations of labeling densities with different sizes of colloidal gold-coated secondary antibodies showed that anti-PIP$_2$ immunolabelings of freeze-fractured membranes were not strongly dependent on gold particle size (Supplementary Fig. 2). This was in line with previous observations of only low steric hindrances of protein-based probes at freeze-fractured membranes[21].

Although being of delicate morphology, cultured neurons grown on sapphire discs can in principle be cryo-preserved and freeze-fractured allowing for immunogold labeling of membrane-associated proteins, such as the F-BAR protein syndapin I[22,23] (PACSIN 1) and the N-Ank protein ankycorbin[24] (RAI14). Primary rat neurons cultured for 16 days (DIV16) on sapphire showed normal postsynaptic morphologies including dendrites, dendritic spines and substructures thereof[22]. We therefore tested whether the combination of cryo-preservation, freeze-fracturing, immunogold labeling and TEM of membrane proteins could be substantially expanded to the detection of the signal lipid PIP$_2$ in its natural, cellular membrane environment (Fig. 1f–k). High power magnifications of freeze-fractured plasma membrane areas of the soma showed frequent anti-PIP$_2$ immunogold labeling (Fig. 1f–f'). The immunogold labeling was found at different membrane topologies and was to a some extend clustered. The clusters of labeling detected usually had a relatively uniform diameter of no more than 100 nm and contained a maximum of 10 labels (Fig. 1f–f').

The labeling was highly specific, as the P-face (protoplasmic face of the fractured plasma membrane) of the membrane representing the cytosolic interface showed an average labeling density of 24.2 gold particles/µm$^2$, whereas intrinsic control surfaces within the specimen showed very low labeling densities (soma E-face (exoplasmic face of the fractured plasma membrane), 3.0/µm;$^2$ ice, 0.6/µm$^2$) (Fig. 1g). These observations are consistent with PIP$_2$ being predominantly present in the inner leaflet of the plasma membrane[25].

Surprisingly, in the same specimen and immunolabelings, the PIP$_2$ labeling density of dendritic membrane areas was much below the very abundant labeling in the somatic plasma membrane (Fig. 1h–k). In dendritic membrane areas, the labeling density only reached 6.3 particles/µm$^2$. It therefore was only roughly a quarter of that of somatic membrane areas.

Importantly, also this much lower dendritic PIP$_2$ detection clearly was specific, as both dendritic E-face and ice surfaces showed

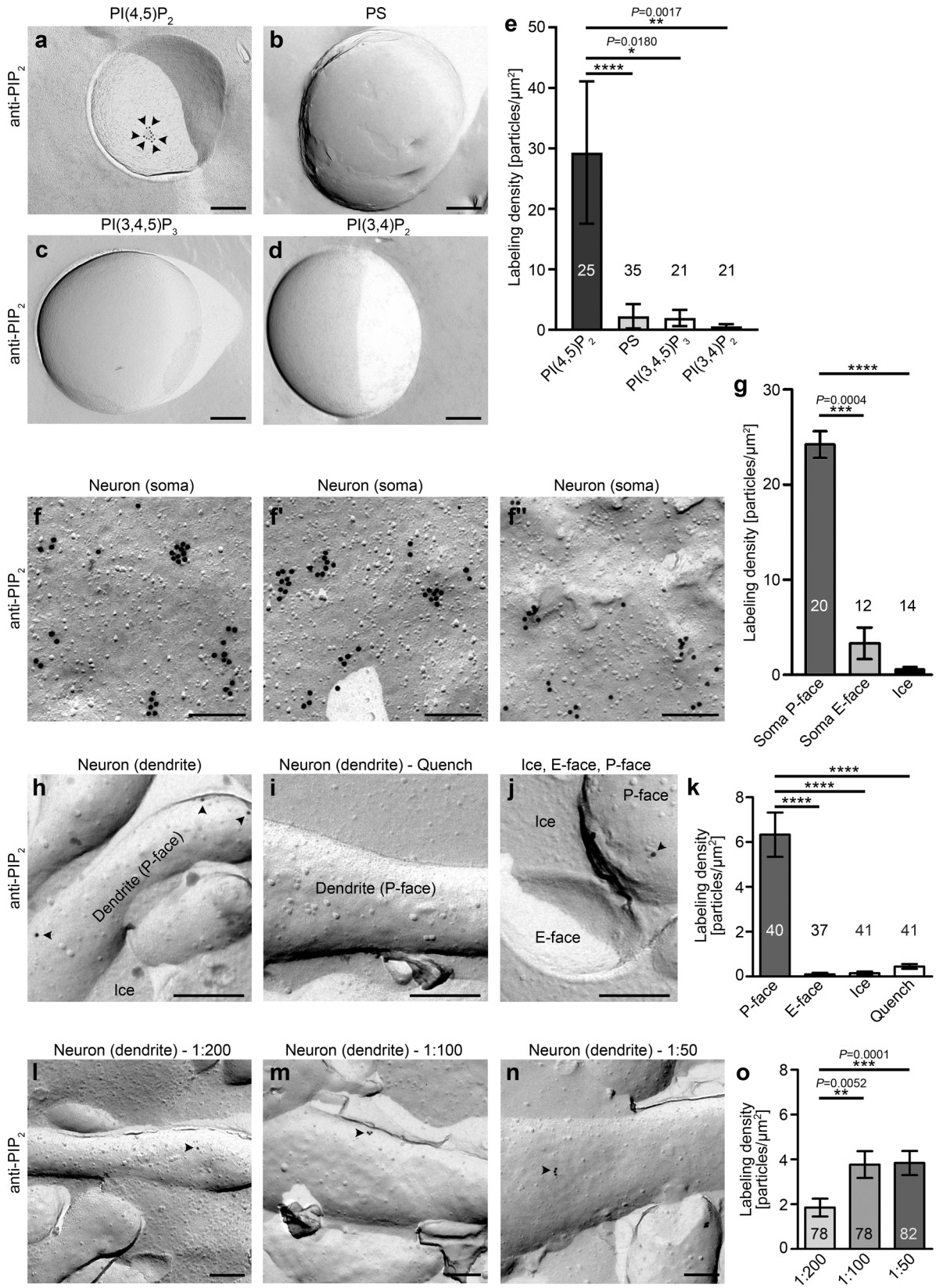

almost no labeling. Furthermore, antibody quenching experiments with PIP$_2$-containing liposomes also resulted in very sparse labeling of freeze-fractured dendritic plasma membranes (Fig. 1i–k).

In order to reliably study the occurrence, intensity, dynamics and decay of PIP$_2$ signals with ultrahigh resolution, it next was important to determine the antibody dilution detecting the maximal amount of PIP$_2$ without putative unspecific binding due to antibody excess. Incubating aliquots of the same freeze-fracture replica with different antibody dilutions we observed that saturation was reached at 1:100 (Fig. 1l–o).

**Fig. 1 Immunogold-labeled PIP$_2$ is specifically detected in freeze-fractured plasma membranes of both soma and dendritic areas of neurons.**
**a** Representative TEM images of anti-PIP$_2$ immunogold-labeled freeze-fracture replica of liposomes composed of 65% (w/v) PE, 30% (w/v) PC and 5% (w/v) of either PI(4,5)P$_2$ (PIP$_2$) (**a**), PS (**b**), PI(3,4,5)P$_3$ (PIP$_3$) (**c**), or PI(3,4)P$_2$ (**d**). Arrowheads highlight examples of gold labels. Bars, 200 nm. **e** Quantitative analyses of labeling densities. PI(4,5)P$_2$, $n = 25$; PS, $n = 35$; PI(3,4,5)P$_3$, $n = 21$; PI(3,4)P$_2$, $n = 21$ (for bar/dot plot presentation of the quantitative data in **e**, see Supplementary Figure 1). **f–o** TEM images (**f, f′, f″, h–j, l–n**) and quantitative analyses (**g, k, o**) of anti-PIP$_2$ immunogold labelings of freeze-fractured plasma membranes of soma (**f, f′, f″, g**) and dendrites (**h–n**) of hippocampal neurons (DIV14-16). Control surface evaluations (E-face, ice) (**g–k**) and control experiments with antibodies quenched for specific binding by preincubation with liposomes containing PIP$_2$ (**i, k**) demonstrated the specificity of labeling. (**l–o**) Additional quantitative experiments with different dilutions of anti-PIP$_2$ antibodies establishing an antibody dilution of 1:100 as yielding saturated PIP$_2$ detection. Bars, 200 nm. Soma P-face, $n = 20$; soma E-face, $n = 12$; ice, $n = 14$ ROIs. Dendrites: P-face, $n = 40$; E-face, $n = 37$; ice, $n = 41$; quench (P-face), $n = 41$ ROIs. Dendrites (P-faces) labeled with different antibody dilutions: 1:200, $n = 78$; 1:100, $n = 78$; 1:50, $n = 82$ ROIs. Data, mean ± SEM of at least two independent assays each. Statistical significance calculations, One-way ANOVA/Tukey´s Multi Comparison test (**g**), Kruskal–Wallis/Dunn's (**e, k, o**). *$P < 0.05$; **$P < 0.01$; ***$P < 0.001$; ****$P < 0.0001$. For $P < 0.0001$, exact $P$-values are not available. Other $P$-values are reported directly in the figure. For numerical source data, see Supplementary Data 1.

**PIP$_2$ mainly localizes to dendritic spine heads.** The validated and optimized anti-PIP$_2$ labeling enabled us to next conduct quantitative ultrahigh resolution examinations of unperturbed PIP$_2$ localization in cryo-preserved and freeze-fractured plasma membrane areas of the dendritic arbor of mature hippocampal neurons. As demonstrated before, it is in principle possible to preserve not only dendrites but also dendritic spines during freeze-fracturing and subsequent sample preparations for electron microscopy and to classify and assign them to the distinct subclasses[22]. We focussed our quantitative analyses on mushroom-type spines, i.e., on spines with a clearly discernible head, because mushroom spines make up the vast majority of spines in the brain, contain mature postsynaptic structures and thus are of utmost physiological importance[1,2] (in the following just referred to as spines). The filigreed dendritic spines protruding from dendrites and their substructures turned out to freeze-fracture with sufficient rates and were anti-PIP$_2$ immunolabeled (Fig. 2a–c). Quantitative analyses showed that the anti-PIP$_2$ immunolabeling density in the systematically sampled spines (i.e., also zero profiles included) was higher ($P = 0.0062$) than the average labeling density of adjacent dendritic areas (Fig. 2d).

Within spines, the spine head showed a strong enrichment of PIP$_2$ over the neck and in particular the base (both $P < 0.0001$; ****). The average anti-PIP$_2$ immunolabeling density in resting spine heads was $6.5 \pm 0.6$ particles/$\mu m^2$ compared to $2.9 \pm 0.7$ particles/$\mu m^2$ and $4.6 \pm 1.3$ particles/$\mu m^2$ in the base and neck, respectively. The labeling density of PIP$_2$ in the spine head plasma membrane thus was more than twice as high as that in spine base areas. (Fig. 2e).

These striking differences revealed that even at steady state, specifically the heads of dendritic spines are prominent areas of PIP$_2$ signals in the dendritic arbor of mature neurons.

**LTD induction leads to transiently increased PIP$_2$ levels in especially the heads of dendritic spines.** Induction of LTD by incubations with 50 μM N-methyl-D-aspartate (NMDA) for up to 3 min and presynaptic silencing (2 μM tetrodotoxin (TTX) preincubation)[26–31] followed by a chase period in preconditioned medium and subsequent quick-freezing at distinct time frames demonstrated that PIP$_2$ can be detected in dendrites as well as in dendritic spines before and after LTD induction (Fig. 3a–h). Quantitative electron microscopical assessments unraveled that, while the PIP$_2$ labeling densities in dendrite areas (even those surrounding dendritic spines) did not significantly change over time (Fig. 3i), the plasma membrane of dendritic spines showed a rapid and highly statistically significant increase of PIP$_2$ density ($P = 0.0049$ (**)) 2 min into NMDA stimulation. PIP$_2$ levels in dendritic spines during this first phase reached 180% of the PIP$_2$ levels prior to LTD induction by NMDA (Fig. 3j).

This first phase was followed by a second, slower increase in PIP$_2$ levels reaching its peak 10 min after the start of LTD induction. Over the total spine, the maximum of this second phase was marked by PIP$_2$ signals that were more than twice as high (220%) as the steady-state PIP$_2$ levels (Fig. 3j). This key finding was subsequently confirmed by additional evaluations by an independent experimenter. Importantly, such efforts led to virtually the same data for spine selection, spine area determinations and labeling counts and therefore yielded also virtually identical data of labeling densities (Supplementary Fig. 3).

Thereafter, PIP$_2$ levels dropped. After 30 min, they nearly attained pretreatment levels. This decline thereby constituted another, a third distinct phase of PIP$_2$ signals upon LTD induction in dendritic spines (Fig. 3j).

Interestingly, more detailed spatial investigations revealed that it was exclusively the spine head membrane but not the membrane areas of the spine base or neck that displayed the increases of PIP$_2$-levels during both phase 1 and phase 2 when normalized to the respective 0 min data (Fig. 3k). In the spine head membrane, the PIP$_2$ labeling density was more than 230% of steady-state levels during phase 1 and more than three times as high during phase 2 (Fig. 3k).

The spine base and the dendritic membrane areas surrounding the spine both showed some trend towards an increase of PIP$_2$ levels indicating some 2D-diffusion of PIP$_2$ out of spine membrane areas during the late phase of LTD induction but both of these trends were minor and remained statistically insignificant (Fig. 3i, k). The overall decline of PIP$_2$ levels in the total spine in phase 3 (Fig. 3j) was predominantly attributable to changes of PIP$_2$ levels in the spine head. Thirty minutes after the start of LTD induction, PIP$_2$ levels in the spine head membranes were back to pretreatment levels (Fig. 3k).

**Different subdomains of the dendritic spine head membrane show distinct behavior and kinetics of PIP$_2$ signals.** The head of the spine undergoes various structural changes during LTD, which are thought to potentially involve different spine subdomains. Measuring the height of each spine head and dividing it into three equally wide zones (Fig. 4a) unveiled that the distribution of PIP$_2$ signals was not equal but showed very distinct behaviors and kinetics in the three head subdomains. A prominent PIP$_2$ increase after NMDA treatment could be seen in the upper head (Fig. 4b). PIP$_2$ signals quickly rose to about 250% of those at steady state during the first phase. They continued to rise to more than 350% during the second phase. Similarly strong increases of PIP$_2$ levels were observed in the middle head area (Fig. 4c; Supplementary Fig. 3k, l).

The kinetics of the upper spine head membrane areas were unique, as they showed a relatively continuous rise of PIP$_2$ levels during phase 1 and 2 (Fig. 4b–d). Furthermore, the drop of PIP$_2$

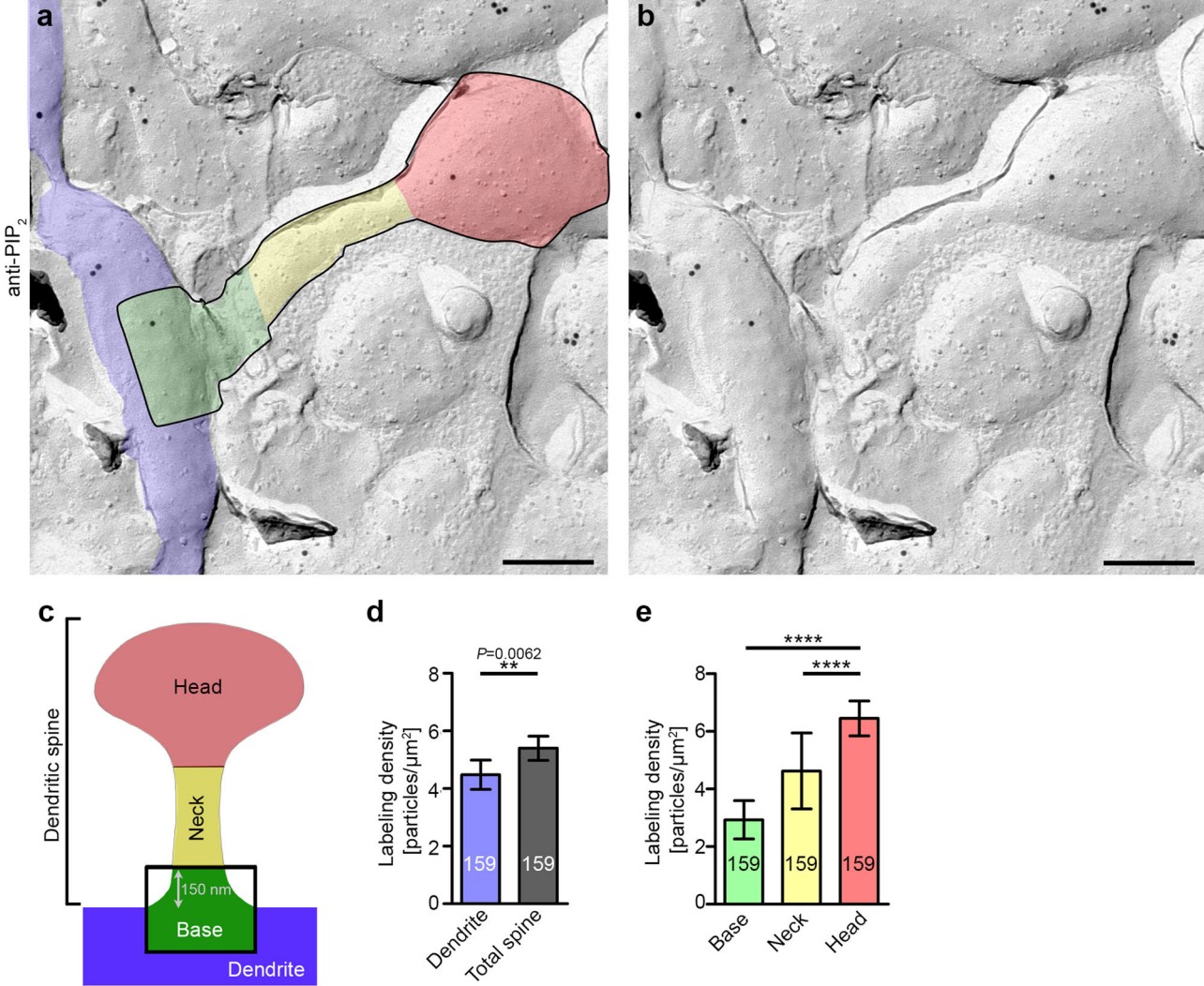

**Fig. 2 At steady state, PIP2 signals are more abundant in dendritic spines than in dendrites and occur in particular in spine heads. a–c** TEM images (**a, b**) of an anti-PIP$_2$ immunogold-labeled dendritic spine of a freeze-fractured hippocampal neuron (DIV16) with outlined spine and transparent coloring of dendrite (blue), spine base (green), spine neck (yellow), and spine head (red) (**a**), as depicted schematically in **c**, and corresponding raw TEM image (**b**). Bars, 200 nm. **d, e** Quantitative analyses of anti-PIP$_2$ immunogold labeling densities of dendritic versus dendritic spine plasma membrane areas (**d**) and of subspine areas (**e**), respectively. Data, mean ± SEM of 16 independent assays; $n = 159$ ROIs each (dendrite, total spine, spine base, neck, and head). Statistical significance calculations, Mann–Whitney (**d**) and Kruskal–Wallis/Dunn's (**e**). **$P < 0.01$; ****$P < 0.0001$. For $P < 0.0001$, exact $P$-values are not available. The other $P$-value is reported in the figure. For numerical source data, see Supplementary Data 1.

levels in phase 3 reached much lower levels in the middle and lower head when compared to data for the upper spine head membrane compartment (Fig. 4b–d).

These observations suggested that LTD-associated PIP$_2$ kinetics differ in the subcompartments of spine heads. The increase in the upper and middle spine head membrane areas may reflect an importance of PIP$_2$ in regulating the availability of AMPA receptors at the PSD and/or the initiation of structural changes in the spine head.

**The initial minutes of LTD induction are marked by PIP$_2$ clusters transiently occurring in spine heads.** It has been suggested that PIP$_2$ is involved in anchoring synaptic AMPA receptors and/or in endocytosis of AMPA receptors[32]. Unfortunately, establishing double-immunogold labelings of AMPA and/or NMDA receptors as well as of different endocytic components, such as clathrin and dynamin, as well as of postsynaptic scaffold proteins, such as PSD95 and ProSAP/Shank proteins, together with the successful PIP$_2$ detection was unsuccessful. In line with

these difficulties, both NMDA and AMPA receptors were detected at the extracellular side after membrane facturing[33–37] and the difficulties to detect proteins, which are merely indirectly or merely peripherally associated to membranes but not inserted, are probably due to the sample preparation procedure, which provides full membrane access by effectively removing such material. Cell biological processes involving PIP$_2$ signaling would presumably rely on some kind of PIP$_2$-enriched nanodomains in spine membranes. Indeed, we were able to observe PIP$_2$ clusters in cryo-preserved, freeze-fractured plasma membranes of cultured neurons (Fig. 5a).

Both the frequency of such clusters as well as the relative fraction of clustered PIP$_2$ was low at steady state but strongly increased when the neurons were stimulated with NMDA. Detailed quantitative analyses of spine heads revealed that merely 22.4% of all anti-PIP$_2$ labels were detected within clusters at steady state (Fig. 5b; 0 min). In contrast, as early as 2 min into the NMDA treatment, 57.6% of all PIP$_2$ labels were found inside of clusters (Fig. 5b; 2 min).

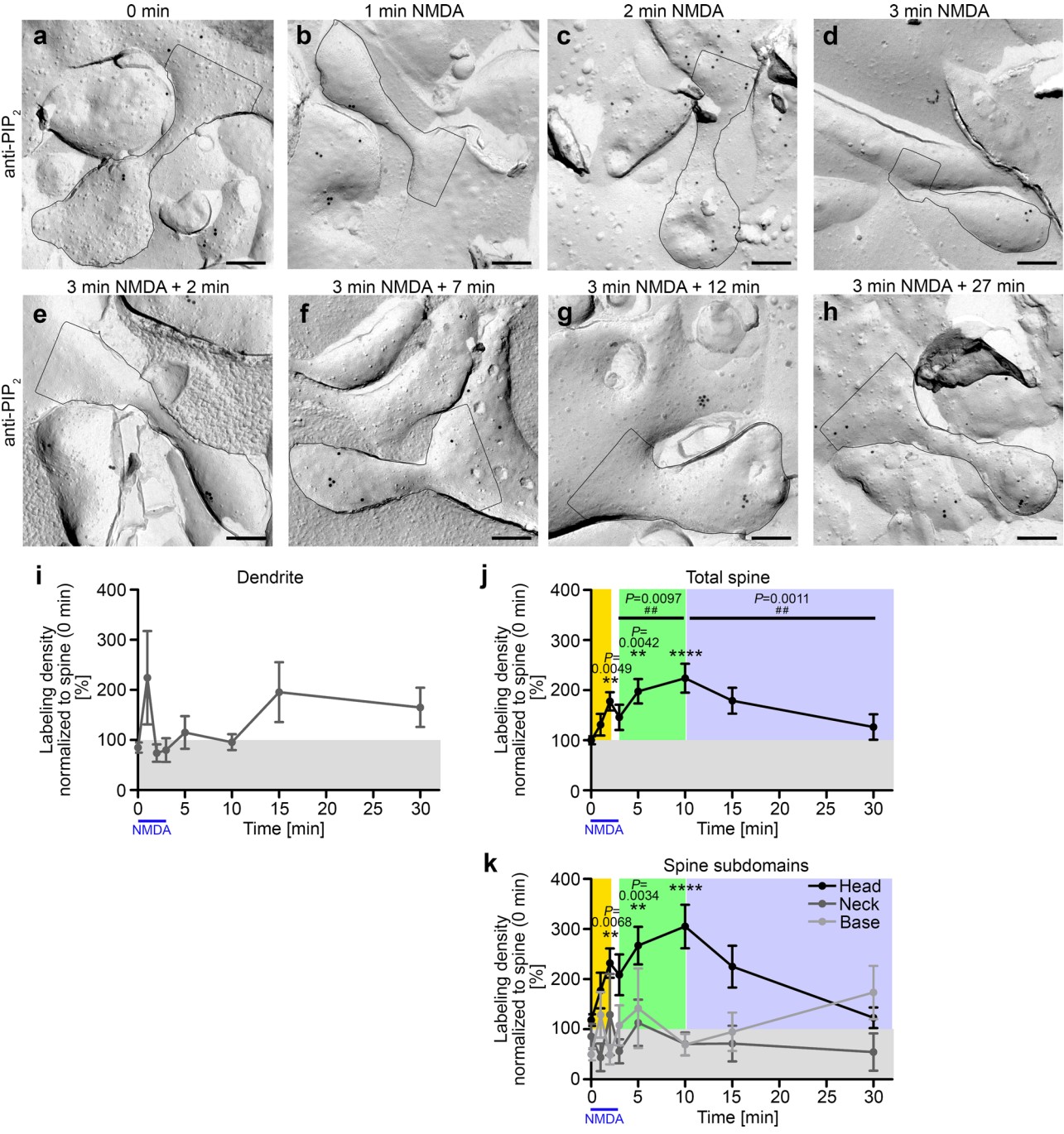

**Fig. 3 PIP2 transiently increases after LTD induction in dendritic spines. a–h** TEM images of anti-PIP$_2$ immunogold-labeled dendritic spines (outlined) of freeze-fractured hippocampal neurons (DIV14-16) at steady state (0 min) and at different times during and after LTD induction by incubation with 50 μM NMDA for 3 min and subsequent placing back into preconditioned medium. Bars, 200 nm. **i–k** Quantitative determinations of anti-PIP$_2$ immunogold labeling densities at different treatment times during and after LTD induction determined in dendrites (**i**), in total spines (**j**) and in the three different spine subdomains (**k**) expressed as percent of the average labeling density of the total spine data at 0 min (steady state) in each assay. Three apparent phases of PIP$_2$ signaling responses in dendritic spines during and after LTD induction are highlighted by coloring (yellow, phase 1, rapid first increase of PIP$_2$ density; green, second phase of increase of PIP$_2$ signals in the spine (head) plasma membrane until a maximum is reached at 10 min; blue, phase 3, subsequent decreasing PIP$_2$ signals in the spine (head)). For a comparison of absolute data for 0 and 3 + 7 min not normalized to the respective controls with corresponding data obtained from an independent, untrained experimenter see Supplementary Fig. 3. (**i–k**) Data, mean ± SEM. 0 min, $n = 159$; 1 min, $n = 36$; 2 min, $n = 53$; 3 min, $n = 49$; 5 (3 + 2) min, $n = 50$; 10 (3 + 7) min, $n = 59$; 15 (3 + 12) min, $n = 44$; 30 (3 + 27) min, $n = 35$ ROIs each (dendrite; total spine; spine subdomains, base, neck, and head) from 3-16 independent assays with different incubation times. Statistical significance calculations, Kruskal–Wallis/Dunn's (**i–k**; **$*P < 0.01$; ****$P < 0.0001$) and Mann–Whitney tests for 3 vs. 10 min and 10 vs. 30 min (**j**; ##$P < 0.01$) (**j**). For $P < 0.0001$, exact $P$-values are not available. Other $P$-values are reported directly in the figure. For numerical source data, see Supplementary Data 1.

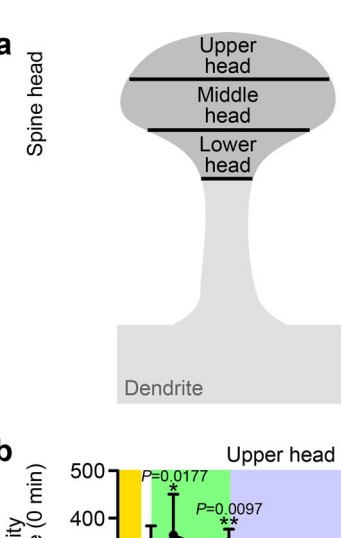

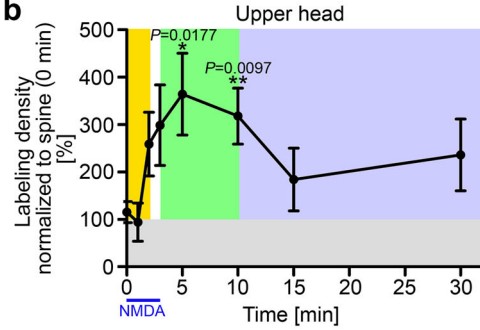

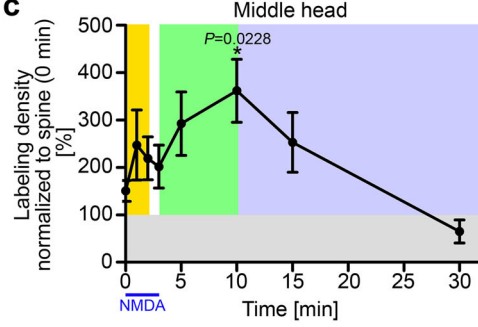

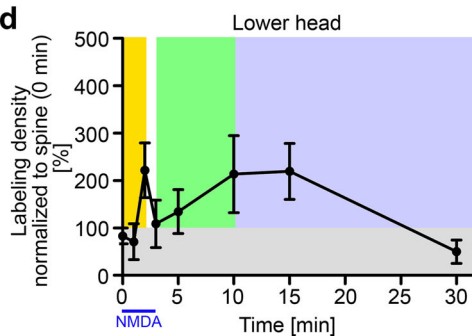

**Fig. 4 Especially upper and middle spine head membrane areas respond to LTD induction by displaying transiently increased PIP2 signals.** The spine heads were further subdivided into lower, middle, and upper head according to the schematic (**a**). Quantitative analyses of the $PIP_2$ labeling densities in DIV14-16 neurons after NMDA treatment (see Fig. 3) in the upper (**b**), middle (**c**), and lower head (**d**). Quantitative analyses are based on the total spine data at 0 min (steady state) in each assay. For a comparison of absolute data for 0 and $3 + 7$ min not normalized to the respective controls with corresponding data obtained from an independent, untrained experimenter see Supplementary Fig. 3. 0 min, $n = 159$; 1 min, $n = 36$; 2 min, $n = 53$; 3 min, $n = 49$; 5 $(3 + 2)$ min, $n = 50$; 10 $(3 + 7)$ min, $n = 59$; 15 $(3 + 12)$ min, $n = 44$; 30 $(3 + 27)$ min, $n = 35$ ROIs each (upper, middle, and lower head) from 3 to 16 independent assays with different incubation times. Data, mean ± SEM. Statistical significance calculations, Kruskal–Wallis/Dunn's. *$P < 0.05$; **$P < 0.01$. P-values are reported directly in the figure. For numerical source data, see Supplementary Data 1.

Also the last phase reflecting the decline of $PIP_2$ levels after LTD induction (15 and 30 min) did not show any statistically significant alteration in $PIP_2$ clustering (Fig. 5b).

The second phase of LTD-induced $PIP_2$ signaling, which is marked by the highest levels of $PIP_2$ in dendritic spine heads (Figs. 3, 4), thus showed a completely different distribution of $PIP_2$ when compared to the initial phase of LTD induction.

**PTEN is involved in the second phase $PIP_2$ increase during LTD but not in the first phase.** Phosphatase and Tensin Homolog deleted on Chromosome 10 (PTEN) has been shown to be recruited to the PSD after LTD induction and it was suggested that $PIP_2$ signals during LTD are generated by $PIP_3$ dephosphorylation[17] (Fig. 6a). Bisperoxovanadium compounds inhibit protein tyrosine phosphatases but especially bpV(HOpic) showed a selectivity for PTEN at low concentrations[38]. To obtain insights into the molecular mechanisms behind the $PIP_2$ accumulations in dendritic spines we observed in the different phases of $PIP_2$ signals during LTD induction, we therefore examined neurons incubated with the PTEN inhibitor bpV(HOpic) for 60 min prior to LTD induction with NMDA (Fig. 6b–j).

At steady state (0 min), treatments with bpV(HOpic) did not result in any significantly different $PIP_2$ levels in dendritic spines compared to pretreatment with solvent control ($ddH_2O$; −bpV(HOpic)) (Fig. 6h). Also the elevated $PIP_2$ levels of the first phase (represented by 2 min NMDA treatment) remained completely unaffected by PTEN inhibition in both total spine plasma membrane areas (Fig. 6i; 235% of steady state; $P = 0.0148$) and in spine head plasma membrane areas (Fig. 6j; 269% of steady state; $P = 0.0321$).

In contrast to the bpV(HOpic)-insensitive first phase of $PIP_2$ dynamics of LTD induction, the second phase of $PIP_2$ dynamics (represented by 3 min NMDA + 7 min) turned out to be affected by PTEN inhibition. While the $PIP_2$ density in spine membranes of neurons not subjected to PTEN inhibition was 246% of steady-state $PIP_2$ levels and thereby was statistically significantly different from the 0 min value ($P = 0.0020$) (Fig. 6i), the $PIP_2$ levels in the plasma membrane of dendritic spines of neurons preincubated with PTEN inhibitor at $3 + 7$ min rather resembled the 0 min data (Fig. 6i).

The inhibitory effects of bpV(HOpic) in the second phase of $PIP_2$ dynamics were especially obvious in spine head membranes. Here, $PIP_2$ levels were strongly elevated in non-inhibitor-treated control neurons at $3 + 7$ min when compared to 0 min (Fig. 6j; 297%; $P = 0.0131$). In contrast, in neurons pretreated with bpV(HOpic), $PIP_2$ levels at $3 + 7$ min were only about 150% of the 0 min data and thereby were statistically significantly different from the about twice as high $3 + 7$ min data obtained in spine

Strikingly, $PIP_2$ clustering mainly seemed to be a short-lived and early phenomenon of the rapid $PIP_2$ increase in spine heads, as the statistically significant and predominant occurrence of $PIP_2$ inside of membrane nanodomains of less than 100 nm in diameter (most were around 50 nm) was observed exclusively 2 min and 3 min after the start of LTD induction (Fig. 5b).

Stunningly, only minutes after conclusion of the NMDA treatment, the now strongly elevated $PIP_2$ in the spine head (compare Fig. 3k) was distributed in a mostly disperse manner, similar to the distribution at steady state (Fig. 5b; compare similar distribution of 0 min vs. $3 + 2$ min and $3 + 7$ min, respectively).

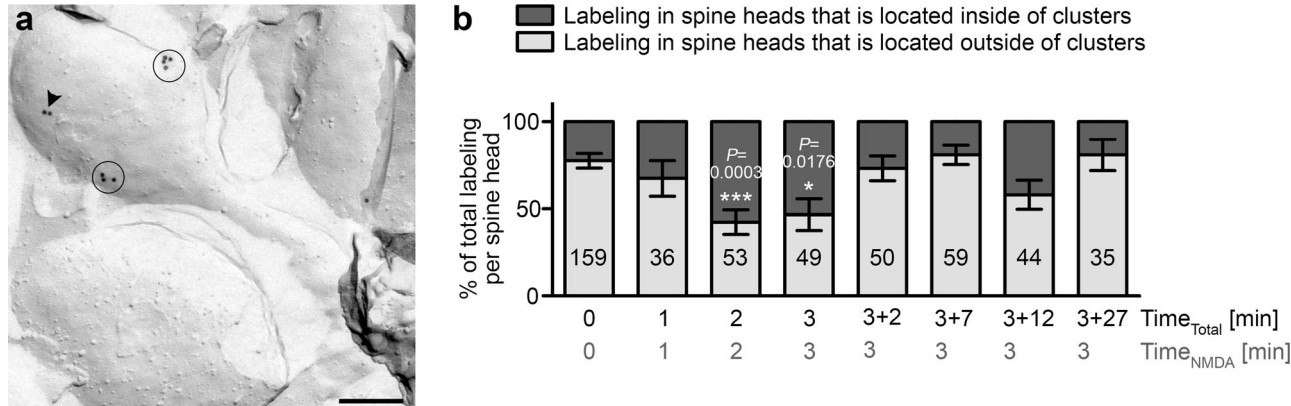

**Fig. 5 The initial minutes of LTD induction are marked by PIP2 clusters transiently occurring in spine heads. a** Example TEM image of a cryo-preserved, freeze-fractured and anti-PIP$_2$ immunogold-labeled spine of a primary hippocampal neuron treated with NMDA for 3 min. Clustered ($n \geq 3$ gold particles in a circular ROI with a diameter of 100 nm) and (more) disperse anti-PIP$_2$ signals in the spine head are marked with circles and an arrowhead, respectively. Bar, 200 nm. **b** Quantification of the distribution of anti-PIP$_2$ immunogold labels in spine heads found within a cluster compared to labels outside of any cluster (in percent of total labeling found at the spine head membrane). Note the transient increase in clustering 2 and 3 min into NMDA treatment. Data, mean ± SEM. 0 min, $n = 159$; 1 min, $n = 36$; 2 min, $n = 53$; 3 min, $n = 49$; 5 (3 + 2) min, $n = 50$; 10 (3 + 7) min, $n = 59$; 15 (3 + 12) min, $n = 44$; 30 (3 + 27) min, $n = 35$ ROIs (spine head) each from 3 to 16 independent assays with different incubation times. Two-way ANOVA/Šídák's Multiple Comparison test (comparisons to 0 min). *$P < 0.05$; ***$P < 0.001$. $P$-values are reported directly in the figure. For numerical source data, see Supplementary Data 1.

head membrane areas of neurons not subjected to PTEN inhibition (Fig. 6j; $P = 0.0259$).

Thus, whereas the initial phase of PIP$_2$ signals in LTD induction is completely independent from PIP$_2$ supplies by PIP$_3$ dephosphorylation, PTEN's enzymatic activity strongly contributes to specifically the second phase of LTD-mediated PIP$_2$ accumulation in dendritic spines heads.

**PIP5K inhibition prevents the LTD-induced PIP$_2$ increase.** In eukaryotes, PIP$_2$ is frequently generated by phosphorylation of PI4P by phosphatidylinositol-4-phosphate-5-kinases (PI4P5Ks/PIP5Ks) (Fig. 7a). Among the various isozymes of PIP5K, PIP5Kγ shows high expression in the brain[39]. UNC3230 selectively inhibited PIP5Kγ[40] in dorsal root ganglion neurons and is the only commercially available inhibitor of PIP5Ks. We therefore used UNC3230 for further dissecting the source of PIP$_2$ during the different phases of PIP$_2$ signals during LTD induction (Fig. 7b–g). Pretreatment of hippocampal neurons for 16 h with 500 nM UNC3230 did not lead to any statistically significant effects on PIP$_2$ levels at steady state (t = 0 min) when compared to solvent control (Fig. 7h).

In contrast, inhibition of PIP5Kγ resulted in a complete block of PIP$_2$ increase during both the first (2 min) and the second phase (3 + 7 min) of PIP$_2$ dynamics during LTD induction. This efficient block of LTD-induced PIP$_2$ accumulation during both phases was underscored by clear and statistically significant differences between the + and the −UNC3230 conditions. The complete impairment of LTD-induced PIP$_2$ dynamics in the dendritic spines was obvious irrespective of whether the total spine (Fig. 7i) or exclusively the membrane areas of the spine head were examined (Fig. 7j). At all +UNC3230 conditions, PIP$_2$ levels remained equal to steady state (0 min) (Fig. 7i, j).

Thus, PIP5Kγ clearly is pivotal in the generation of PIP$_2$ during NMDA-induced LTD induction and, in contrast to the phase-2-selective contribution of PTEN, PIP5Kγ's is absolutely critical during both of the initial phases of PIP$_2$ signals during LTD induction identified in our quantitative analyses.

**PLC inhibition prevents the decline of PIP$_2$ levels towards levels resembling steady state.** In order to understand LTD-induced PIP$_2$ dynamics in dendritic spines, it was next

important to investigate the molecular mechanisms bringing about the effective removal of PIP$_2$ from dendritic spine plasma membrane areas in phase three of PIP$_2$ dynamics during LTD induction. Besides diffusion of PIP$_2$ from spine head areas, several enzymes catalyze reactions converting PIP$_2$. Most prominently, phospholipase C (PLC) cleaves PIP$_2$ into diacylglycerine (DAG) und inositol-1,4,5-trisphosphate (IP$_3$) (Fig. 8a). We therefore treated primary hippocampal neurons with the PLC inhibitor U-73122[41] in comparison to the solvent control (dimethylsulfoxide (DMSO)) and examined PIP$_2$ levels with and without LTD induction (Fig. 8b–j). U-73122 did neither lead to any alterations of PIP$_2$ labeling densities prior to LTD induction (Fig. 8h) nor did it alter the rapid increase of PIP$_2$ levels during the initial phases of LTD induction, as even after 10 min (3 + 7 min) PIP$_2$ levels remained similar to those of dendritic spines and dendritic spine heads not treated with inhibitor, respectively (Fig. 8i, j).

In contrast, the third phase of PIP$_2$ signaling during LTD induction turned out to be completely dependent on PLC activity. Whereas after 30 min, control dendritic spine plasma membrane areas and control spine head membrane areas displayed a complete reversal of the transient, LTD-induced PIP$_2$ increase, PLC inhibition with U-73122 completely blocked this decline. At 30 min, the anti-PIP$_2$ labeling densities in both total spine plasma membrane areas as well as in spine head membrane areas persisted at highly elevated levels (Fig. 8i, j).

These results clearly unveiled that LTD-induced PIP$_2$ signals in the dendritic spine head membrane are actively terminated by PIP$_2$ degradation by PLC. Altogether, our data thereby unveil a transient increase of PIP$_2$ signals in the plasma membrane of dendritic spines, highlights the kinetics of these signals and unravels molecular mechanisms that play crucial roles in the dynamics of LTD-induced PIP$_2$ signals in spine head membrane areas.

**Discussion**
Signaling cues emanating from phosphoinositides are considered as key aspects for a variety of cellular functions, yet, these temporal and spatial cues are very difficult to visualize and study in cellular membranes. Here, we describe the to our knowledge first ultrahigh resolution immunodetection of PIP$_2$. Our analyses

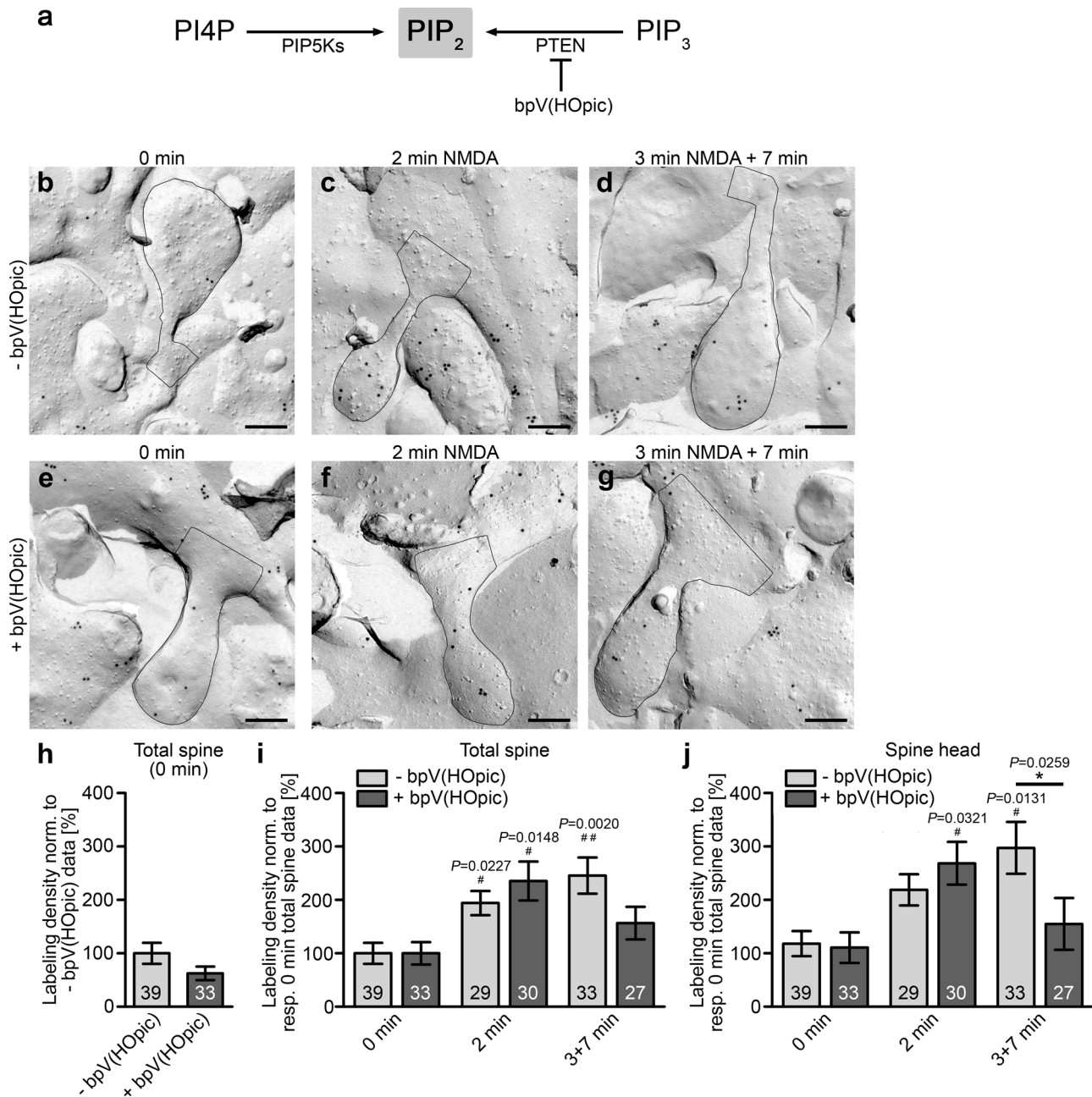

**Fig. 6 The second phase of LTD-associated PIP2 dynamics is dependent on PTEN in the spine head whereas the early phase of LTD induction is not.**
**a** Scheme visualizing suggested routes towards PIP$_2$ during LTD and the inhibition by bpV(HOpic). **b–g** Representative TEM images of cryo-preserved, freeze-fractured and anti-PIP$_2$ immunogold-labeled spines of DIV14-16 hippocampal neurons pretreated for 60 min with ddH$_2$O as a vehicle control (**b**-**d**) or with 15 nM bpV(HOpic) (**e**-**g**) and subjected to NMDA-induced LTD for 2 min (**c**, **f**) and for 3 min NMDA followed by 7 min postincubation time (**d**, **g**) in comparison to non-NMDA-treated controls (0 min) (**b**, **e**). Bars, 200 nm. **h** Quantitative analyses of the anti-PIP$_2$ labeling density in neurons pretreated with bpV(HOpic) compared to the merely ddH$_2$O-treated neurons (−bpV(HOpic)) in the same assays not revealing any statistically significant effect of PTEN inhibition on basal PIP$_2$ levels. The labeling densities were normalized to the respective - bpV(HOpic) control. **i**, **j** Quantitative analyses of PIP$_2$ signaling during the first and second phase of LTD induction in the total spine (**i**) and in the spine head (**j**), respectively, (both normalized to the respective 0 min data for the total spine labeling densities of each of the two conditions, i.e., with and without inhibitor). Data, mean ± SEM. −bpV(HOpic), 0 min, $n = 39$; 2 min, $n = 29$; 10 (3 + 7) min, $n = 33$ ROIs each (total spines; spine heads) and +bpV(HOpic), 0 min, $n = 33$; 2 min, $n = 30$; 10 (3 + 7) min, $n = 27$ ROIs each (total spines; spine heads) of primary neurons from 3 independent assays. Mann–Whitney (**h**); Two-way ANOVA/Bonferroni's Multiple Comparison between ±bpV(HOpic) conditions (**i**, **j**). *$P < 0.05$. Additional Kruskal–Wallis/Dunn's Multiple Comparisons were conducted for comparison of + and −bpV(HOpic) data at 2 min and at 3 + 7 min to 0 min control data (**i**, **j**) #$P < 0.05$; ##$P < 0.01$. $P$-values are reported directly in the figure. For numerical source data, see Supplementary Data 1.

provide the quantitative spatial and temporal information on PIP$_2$ in the plasma membrane of the postsynaptic compartment of neuronal cells that is required to follow PIP$_2$ signals in synaptic plasticity. Surprisingly, compared to the PIP$_2$ levels found in

plasma membrane areas of the soma of neurons, which were in line with abundant PIP$_2$ detections using purified GST-fusion proteins of the PH domain of PLC in non-neuronal cells[21,42,43], the PIP$_2$ levels in the plasma membrane of the dendritic tree

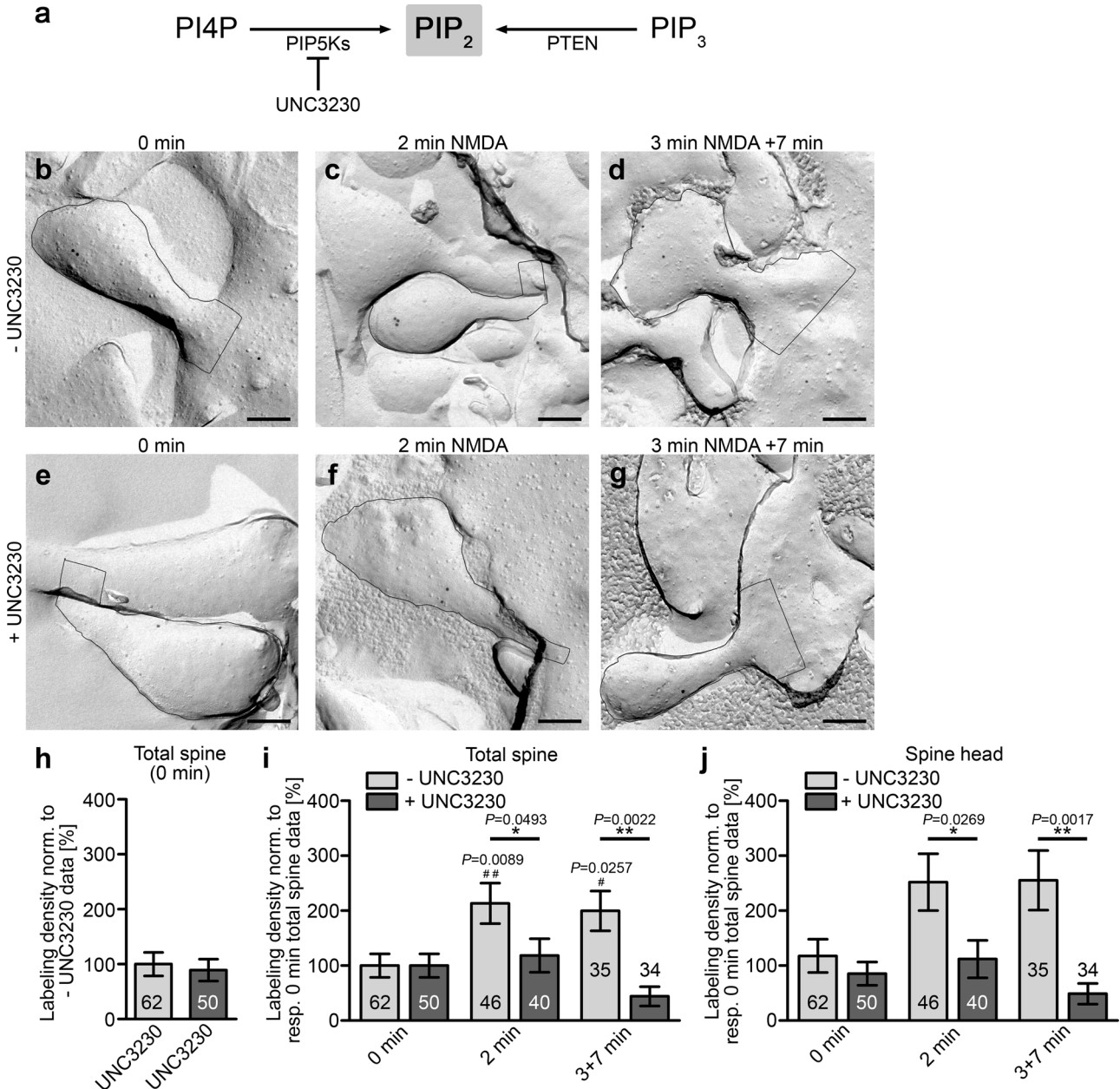

**Fig. 7 PIP5K inhibition prevents the PIP2 increase during both the first and the second phase of LTD induction. a** Scheme visualizing suggested routes towards $PIP_2$ during LTD and the inhibition by the PIP5K inhibitor UNC3230. **b–g** TEM images of dendritic spines of anti-$PIP_2$ immunolabeled, freeze-fractured spines of DIV14-16 hippocampal neurons, which were pretreated with vehicle control (0.002% DMSO) (−UNC3230, **b–d**) or with 500 nM UNC3230 (**e–g**) prior to either inducing LTD by 2 min NMDA (50 μM) or with 3 min NMDA and 7 min recovery time (3 + 7 min), respectively, or cryopreserving the cells at steady state (0 min). Bars, 200 nm. **h** Quantitative analyses of the anti-$PIP_2$ labeling density of +UNC3230 and −UNC3230 neurons at steady state normalized to the respective untreated control. **i, j** Quantitative analyses of $PIP_2$ dynamics revealing a very strong negative impact of PIP5K inhibition on both the first (2 min) and the second (3 + 7 min) phase of elevated $PIP_2$ signals during LTD induction. Data, mean ± SEM. −UNC3230, 0 min, $n = 62$; 2 min, $n = 46$; 10 (3 + 7) min, $n = 35$ ROIs each (total spines; spine heads) and + UNC3230, 0 min, $n = 50$; 2 min, $n = 40$; 10 (3 + 7) min, $n = 34$ ROIs each (total spines; spine heads) of primary neurons from 3 independent assays. Mann–Whitney (**h**; n.s.); Two-way ANOVA/ Bonferroni's Multiple Comparison between ±UNC3230 conditions (**i, j**) *$P < 0.05$; **$P < 0.01$. Additional Kruskal–Wallis/Dunn's Multiple Comparison for comparison of +UNC3230 and −UNC3230 data at 2 min and 3 + 7 min to 0 min control data (**i, j**)#.$P < 0.05$; ##$P < 0.01$. P-values are reported directly in the figure. For numerical source data, see Supplementary Data 1.

generally were very low. In the dendritic compartment, $PIP_2$ thus appears to act as bona fide signaling cue.

Using light microscopy and the $PIP_2$-quenching PH domain of PLC as a protein probe, Horne and Dell'Acqua[16] suggested that $PIP_2$ is strongly enriched in dendritic spines in comparison to dendrites. Our quantitative determinations showed that, by using freeze-fracturing methods, this enrichment of $PIP_2$ in dendritic

spines at steady state in fact is small. Our electron microscopic examinations unveiled that exclusively spine head membranes showed a strong enrichment (about +40%) over general dendrite levels.

Although super-resolution methods have been applied for the detection of $PIP_2$ in PC12 cells[44,45], light microscopy is of limited use for detailed correlations with substructures of dendritic spines

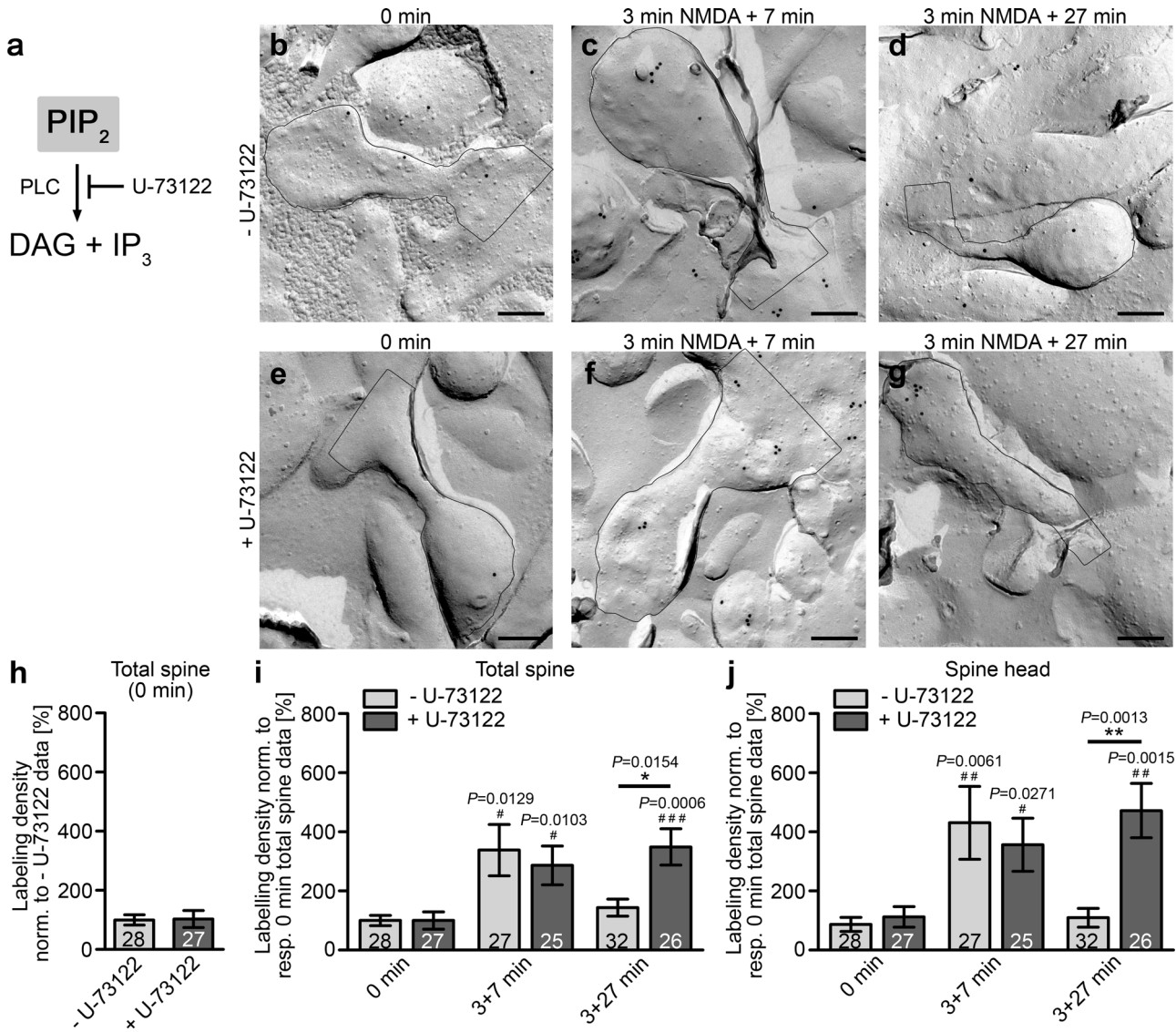

**Fig. 8 Inhibition pf PLC prevents the return of PIP2 levels to steady-state levels in phase 3 of LTD induction. a** Scheme visualizing the degradation of $PIP_2$ yielding $DAG + IP_3$ and the inhibition by the PLC inhibitor U-73122. **b–g** TEM images of dendritic spines of anti-$PIP_2$ immunolabeled, freeze-fractured spines of DIV14-16 hippocampal neurons, which were merely pretreated with vehicle control (0.2% DMSO) (−U-73122, **b–d**) or with 10 µM U-73122 (**e–g**) prior to either inducing LTD by 3 min NMDA and 7 min recovery time (3 + 7 min) and by 3 min NMDA and 27 min recovery time (3 + 27 min), respectively, or leaving the cells at steady state (0 min). Bars, 200 nm. **h** Quantitative analyses of the anti-$PIP_2$ labeling density of + U-72122 and −U-73122 neurons at steady state normalized to the respective untreated control. (**i**, **j**) Quantitative analyses of $PIP_2$ dynamics demonstrating that PLC inhibition has no effect on $PIP_2$ levels during the second phase (3 + 7 min) of LTD induction but completely blocks the restauration of steady-state $PIP_2$ levels during the third phase (3 + 27 min) of $PIP_2$ dynamics during LTD induction. Data, mean ± SEM. −U-73122, 0 min, $n = 28$; 3 + 7 min, $n = 27$; 3 + 27 min, $n = 32$ ROIs each (total spines; spine heads) and +U-73122, 0 min, $n = 27$; 3 + 7 min, $n = 25$; 3 + 27 min, $n = 26$ ROIs each (total spines; spine heads) of primary neurons from 3 independent assays. Mann–Whitney (**h**; n.s.); Two-way ANOVA/Bonferroni's Multiple Comparison between ±U-73122 conditions (**i**, **j**). *$P < 0.05$; **$P < 0.01$. Additional Mann–Whitney tests for comparisons of +U-73122 and −U-73122 data at 3 + 7 min and 3 + 27 min to 0 min control data (**i**, **j**). #$P < 0.05$; ##$P < 0.01$; ###$P < 0.001$. P-values are reported directly in the figure. For numerical source data, see Supplementary Data 1.

in neuronal cells, as these substructures are very small. Importantly, exogenous expression of reporter proteins, such as PLC-PH, although being widely used as $PIP_2$ probe, furthermore does not unambiguously report $PIP_2$ level changes because PLC-PH has been shown to be displaced from $PIP_2$ by the increase of $PIP_2$'s intracellular degradation product $IP_3$[6,46]. This limitation is severe, as the half-live of $PIP_2$ in cells is only about 1 min[47,48]. PLC-PH expression furthermore impacts cell physiology and signaling by competing with the $PIP_2$-degrading enzyme PLC and by quenching $PIP_2$ signals[49,50], as PLC-PH bound to $PIP_2$ will compete with the binding of $PIP_2$ effector proteins, such as

endocytic, cytoskeletal and signaling components[7] and thereby hamper proper recognition and transmission of $PIP_2$ signals. Importantly, also chemical fixation procedures are prone to interfere with the physiological distribution of lipid signals[50,51]. The combination of very rapid cryo-preservation, freeze-fracturing, irreversibly stabilizing membrane components by carbon and platinum evaporation, immunogold labeling of the membrane replica and TEM examination overcomes all of these former limitations and provides a clear view at $PIP_2$ signals during LTD.

Our quantitative assessments of relative $PIP_2$ levels, distribution and organization and the different inhibitor studies unveil

that PIP$_2$ signaling during LTD occurs in a defined pattern. First, a rapid increase of PIP$_2$ levels was observed within the first 2 min of LTD induction by NMDA application. In membrane areas of spine heads, this first increase reached almost 250% of the steady-state levels of PIP$_2$. A role of PIP$_2$ in LTD induction is in line with the finding that chemically induced dimerization to translocate the PIP$_2$-degrading enzyme inositol polyphosphate 5-phosphatase to the plasma membrane and to thereby deplete PIP$_2$ at the plasma membrane led to a disruption of LTD induction by low-frequency stimulation[19]. The observed increase in PIP$_2$ during early LTD induction we determined stands in sharp contrast to observations of a drop in PIP$_2$ concentration in dendritic spines 1 min after NMDA stimulation[16]. The respective conflicting data, however, were obtained by using fluorescence microscopy and overexpression of GFP-tagged PLC-PH and thus were affected by all technical and scientific limitations already discussed above. The strong increase of PIP$_2$ levels in the plasma membrane of dendritic spines we observed in our ultrahigh resolution studies of freeze-fractured membranes was specifically localized to plasma membrane areas of the spine head. This rapid increase of PIP$_2$ in the spine head is in line with a role of PIP$_2$ in the induction of LTD, as LTD is thought to mostly involve receptor and actin cytoskeletal reorganizations in spine heads[1–4].

Strikingly, the initial minutes of PIP$_2$ signals during LTD also were the only time with a statistically significant clustering of PIP$_2$. PIP$_2$ clusters have been observed in clathrin-mediated endocytosis[32]. PIP$_2$ binds to various proteins involved in endocytosis and is essential for initiating and sustaining the assembly of endocytic coated pits[52–59]. PIP$_2$ clustering at nascent endocytic sites may recruit and increase the local concentration of proteins, such as epsin and BAR-domain containing proteins, that in turn contribute to membrane deformation and endocytic vesicle formation[60,61].

NMDA-mediated LTD induction is marked by AMPA receptor endocytosis in the first minutes after stimulation[31,62,63] and this internalization is thought to preferentially occur in perisynaptic areas[63]. These observations are in line with both the time frame of PIP$_2$ signals in dendritic spines we determined and with the accumulation and perseverance of strongly elevated PIP$_2$ levels in specifically the upper and the middle plasma membrane areas of dendritic spines heads after LTD induction observed in our more detailed analyses of spine membrane subdomains.

Our inhibitor studies indicated that the rapid PIP$_2$ increase during initial LTD induction is almost exclusively generated from PI4P. In line with this, PIP5K$\gamma$ is thought to be mostly responsible for synthesis of PIP$_2$ at synapses[64]. Interestingly, dephosphorylated PIP5K$\gamma$661 associates with the endocytic adaptor complex AP2[65,66] and neurons expressing a kinase-dead PIP5K$\gamma$661 mutant or a mutant unable to associate with the AP2 complex did not exhibit LTD after LFS[20].

The second phase of PIP$_2$ signals in dendritic spines during LTD induction was marked by a further but less rapid increase of PIP$_2$ levels, which was limited to specifically the upper and middle plasma membrane areas of the spine head. Again, PIP5K inhibition by UNC3230 abrogated this time period of PIP$_2$ dynamics. Additionally, PIP$_2$ generation from PIP$_3$ contributed to the elevated PIP$_2$ levels of this phase, as demonstrated by PTEN inhibition using bpV(HOpic). This finding is in line with the observations that PTEN accumulated at the PSD after NMDA incubation and inhibition of PTEN by bpV(HOpic) abolished LTD in hippocampal neurons[17]. In contrast, the first phase of PIP$_2$ dynamics was insensitive to PTEN inhibition. These results clearly highlight the biphasic nature of the increase of PIP$_2$ levels observed in dendritic spine heads during LTD induction.

Temporal signaling events require timely termination. This is the major aspect of the third phase of PIP$_2$ dynamics starting about 10 min after NMDA-induced LTD induction. Steady-state PIP$_2$ levels in dendritic spines were observable after about 30 min. It is important to note that, in contrast to using overexpressed PLC-PH as PIP$_2$ probe, the direct antibody-based detection of PIP$_2$ we applied for our analyses is not affected by dissociation from PIP$_2$ due to IP$_3$ increase. The observed decline in labeling density thus truly reflects a decline of PIP$_2$ in the plasma membrane of dendritic spines. While trends of slightly elevated PIP$_2$ levels in the plasma membrane areas of the spine base and the surrounding dendrite may suggest some contribution of PIP$_2$ diffusion out of the PIP$_2$-enriched spine head membrane areas, the observed decline was clearly brought about by an active degradation of PIP$_2$ signals by PLC, as demonstrated by PLC inhibition with U-73122. Consistent with some importance of PLC in LTD, application of U-73122 was described to impair LTD[67].

It seems conceivable that the observed reduction in PIP$_2$ levels at later stages is related to changes of the actin cytoskeleton, which eventually lead to the shrinkage of dendritic spine heads observed upon LTD. PIP$_2$ may oppose the required actin reorganizations and stabilize actin filaments because it for example inhibits the F-actin severing activity of cofilin[68] and gelsolin[69]. PIP$_2$ may furthermore oppose spine shrinkage by linking the actin cytoskeleton to the plasma membrane[70–73]. Our observation that transiently elevated PIP$_2$ levels reached steady-state levels in dendritic spine head membranes 30 min after LTD induction is temporally very well in line with the observation that, following low-frequency stimulations of hippocampal slices, reductions in spine head volume and diameter could be observed 15–60 min after stimulation[74].

Altogether, this study provides ultrahigh-resolution views of PIP$_2$ during LTD induction and unravels the spatial and temporal principles and the molecular mechanisms underlying PIP$_2$ signals and their timely termination during LTD induction in subdomains of the dendritic spine membrane.

## Methods

**Liposome preparation.** Liposomes were essentially prepared according to Reeves and Dowben[75]. In detail, mixtures of lipids in chloroform and methanol (98:2 (v/v)) were spread thinly on the bottom of a Fernbach flask. The mixtures consisted of 65% (w/v) phosphatidylethanolamine (PE), 30% (w/v) phosphatidylcholine (PC) and 5% (w/v) PIP$_2$, PIP$_3$, PI(3,4)P$_2$ and PS, respectively. Lipids were dried under a steady flow of nitrogen gas for 30–60 min and any remaining solvents were removed by a 1 h incubation in a dessicator. Next, the lipids were rehydrated with a flow of water-saturated nitrogen for 30–60 min before adding 5 ml 0.3 M sucrose in ddH$_2$O to allow further hydration and detachment of liposomes from the flask for 14 h at 37 °C. The suspension containing the liposomes was then centrifuged at 200,000 × $g$ for 1 h at 28 °C. The pellet containing the liposomes was resuspended in 20 µl of 0.3 M sucrose in ddH$_2$O with a final lipid concentration of 8 mg/ml.

**Animals.** The rats (Crl:WI; Charles River) used to obtain biological material were bred by the animal facility of the Jena University Hospital in strict compliance with the EU guidelines for animal experiments (approved by the Thüringer Landesamt, Bad Langensalza; Germany). As exclusively primary cells were prepared from post-mortem pregnant rats, no permission for animal experiments was required for this study. Embryos used for preparations of primary hippocampal neurons were E18 and of (undetermined) mixed sex.

**Isolation of primary rat hippocampal neurons and cell culture.** NIH3T3 cells were maintained and cultured under standard conditions.

Rat (Crl:WI; Charles River) primary hippocampal neurons were prepared and cultured largely according to principles developed by Banker and Cowan[76] and established previously[23]. In detail, brains of E18 rats were dissected in ice-cold HBSS (Invitrogen). Isolated hippocampi were then trypsinized with 0.05% trypsin/EDTA (Invitrogen) for 15 min at 37 °C. The supernatant was exchanged for Neurobasal medium (Invitrogen) and supplemented with B-27 supplement (Invitrogen) and 0.5 mM L-glutamate. The tissue was then dissociated into single cells by careful trituration with a 1 ml Pasteur pipette.

As a preparation for freeze-fracturing, neurons needed to be cultured in 24-well plates with sapphire discs (Rudolf Brügger, Swiss Micro Technology)[22,23]. As a preparation, sapphire discs were carefully cleaned and washed and then coated

with poly-D-lysine overnight at 37 °C. The poly-D-lysine was aspirated and carefully removed by washing three times for 10 min with ddH$_2$O. Sapphire discs were then dried and irradiated with ultraviolet light for 15 min.

Primary neurons were seeded at a density of 80,000 cells per well into a 24-well plate supplemented with the prepared sapphire discs. The primary hippocampal neurons were then cultured in Neurobasal$^{TM}$ medium containing 2 mM L-glutamine, 1× B27, and penicillin/streptomycin (100 U and 100 µg, respectively, per ml) for about 2 weeks (at 37 °C, 90% humidity and 5% CO$_2$). At DIV14-16 on sapphire, neurons had formed synapses and neuronal networks[22] and were then used for the experiments.

**Chemical LTD induction in hippocampal neurons and inhibitor studies**. LTD was induced with NMDA[26,31]. In detail, hippocampal neurons were presynaptically silenced by a preincubation with 2 µM TTX (60 min) before inducing LTD with 50 µM NMDA for up to 3 min followed by chase in preconditioned medium for distinct time points of analysis.

To investigate the pathways transiently providing PIP$_2$ during LTD induction and leading to a decline of PIP$_2$ levels at subsequent time points, inhibitors or their corresponding solvent control were added before the NMDA treatment. Final concentrations and preincubation times were as follow: bpV(HOpic), 15 nM in ddH$_2$O for 60 min; U-73122, 10 µM for 60 min in 0.2% (v/v) DMSO; UNC3230 at 500 nM for 16 h in 0.002% (v/v) DMSO.

**Freeze-fracturing and immunolabeling of freeze-fracture replica**. Copper profiles (0.6 mm high) were used for freeze-fracturing. Prior to their use, the profiles were cleaned in a sonication bath with 4% (w/v) tartaric acid, washed in pure acetone and then stored in pure methanol.

For freeze-fracturing of liposomes, 2 µl of the liposome solution was distributed between two copper profiles using the sandwich double-replica technique[77]. The sandwich was immediately plunge-frozen in liquid propane/ethane (1:1) cooled in liquid nitrogen (cooling rate >4000 K/s)[78].

Three sandwiches were placed in a double-replica specimen table also cooled in liquid nitrogen. The samples were then transferred into a BAF400T freeze-fracture machine (Leica) cooled to −140 °C. After the vacuum was established and the pressure was no more than 10$^{-6}$ mbar, the table was flipped open so that the two sandwiched profiles with the sample between them separated and freeze-fractured the sample. Immediately, a carbon coat of 15–20 nm was evaporated onto the samples at a 90° angle, followed by an about 2 nm platinum/carbon coat at a 35° angle[78]. The samples were extracted from the freeze-fracture machine, thawed, floated onto 2.5% (w/v) SDS and incubated under gentle shaking overnight at RT.

After three times 10 min washing in PBS, any unspecific binding was blocked by incubation in 1% (w/v) BSA, 0.5% (w/v) fish gelatine, 0.005% (v/v) Tween® 20 in PBS (pH 7.2) (LBB) for 30 min at RT. The replica were then incubated overnight at 4 °C with mouse monoclonal anti-PIP$_2$ antibodies (Enzo Life Sciences) in LBB (standard condition, 1:100, i.e., 10 µg/ml). Unbound primary antibody was removed by washing three times with LBB (10 min each). The samples were then incubated for 2 h in gold particle-bound goat anti-mouse secondary antibodies diluted in LBB (at RT) (5, 10, and 15 nm gold; BBI Solutions). Unbound secondary antibody was removed by washing (3 × 10 min) with PBS. The immunolabeled replica were then fixed with 0.5% (v/v) glutaraldehyde in PBS for 10 min, washed twice for 10 min with ddH$_2$O, mounted on uncoated copper grids and dried.

For freeze-fracturing of NIH3T3 cells[77,78], a suspension of the cells was placed between two copper profiles similar as described above for analyses of liposomes and then freeze-fractured and immunolabeled as described above.

For neuronal freeze-fracture experiments[22-24], neurons were grown on poly-D-lysine coated sapphire discs (see above). The sapphire discs were placed onto 0.8 mm high copper profiles to fit into the double-replica specimen table. A droplet of 20% (w/v) BSA was added in between the sandwich parts to avoid freezing artefacts. Freeze-fracturing and immunolabeling was then conducted as described above.

**Transmission electron microscopy**. Replica were examined with a EM902A transmission electron microscope (Zeiss) operated at 80 keV.

Imaging was done using a 1 k FastScan CCD camera (TVIPS camera and software). Images were digitalized with EM-Menu 4 software (TVIPS)[22-24].

**Evaluation of antibody specificity and establishment of labeling conditions of freeze-fractured samples by determinations of labeling densities**. Detectability of PIP$_2$ in membranes by anti-PIP$_2$ antibodies was first addressed using PIP$_2$-containing liposomes. Liposomes containing PIP$_3$, PI(3,4)P$_2$, and PS, respectively, instead of PIP$_2$ were examined as controls. Images were taken by randomized sampling, i.e., irrespective of PIP$_2$ labeling and including zero profiles. Same applies to imaging of NIH3T3 cell membranes.

Anti-PIP$_2$ immunolabeling specificity determinations with freeze-fractured membranes of neurons were done with rat hippocampal neurons at DIV14-16 and systematic sampling across the grids. Areas of all three categories (ice and E-face as controls versus P-face) were measured, the amount of gold particles per area was counted and the labeling densities (determined as particles per µm$^2$) were compared to test for antibody specificity.

Secondary antibody controls and freeze-fractured samples, for which primary antibody binding was quenched by preincubation of the antibody with liposomes with added PIP$_2$, served as additional controls.

Additional quantitative experiments with different dilutions of anti-PIP$_2$ antibodies (1:200, 1:100, 1:50 corresponding to concentrations of 5, 10, and 20 µg/ml) were used to establish an antibody dilution yielding a saturated PIP$_2$ detection.

**Analysis of PIP$_2$ immunogold labeling densities and of PIP$_2$ distribution in hippocampal neurons**. Prior to visualization by TEM, the samples were blinded by a colleague. Blinding of recorded images was done using the Ant Renamer software (antp.be/software/renamer). TEM images were obtained by systematic screening of the grids.

Images of freeze-fractured plasma membrane areas of the cell soma and of dendritic arbor were recorded by systematic recording, the images were blinded (see above), the respective areas (E-face, P-face, ice) were determined and the anti-PIP$_2$ labeling density was determined. All morphological structures that could clearly be categorized as mushroom spines as well as neighboring dendrite segments not decorated with spines were used for the analysis, i.e., only images with dendritic spines smaller than 0.75 µm and larger than 2 µm in length and images of spines without any visible dendritic attachment were discarded.

Spines were subdivided into base, neck and spine head. For some analyses, the spine head was further broken down into three parts by dividing the length of the head in three parts equal in height resulting in an upper, middle and lower head. The procedure for area definitions were as follows: First the base was defined. A ground line was drawn interconnecting dendritic membrane points adjacent to the spine. Two further parallel lines of equal length flanking the ground line were drawn at distances of 150 nm and then connected (90° angles). The included cellular areas were defined as base area (see also Fig. 2). From a position representing half of the ground line another line was drawn to the tip of the spine to define its longitudinal axis. Rectangular from the longitudinal spine axis, a line was drawn in a 90° angle to separate the neck from the head region (Fig. 2). This border usually was easily recognizable by the 2D widening of the neck and by the 3D information provided by the platinum shadowing (Fig. 2). The head area was further split into three areas (lower head, middle head and upper head) by two lines rectangular to the longitudinal axis drawn at positions reflecting one third and two thirds of the head length defined by the longitudinal axis (Fig. 4a). All areas were then surrounded using the polygon selection tool in ImageJ/FIJI following the predefined border lines and the membrane profile, respectively, and measured using ImageJ.

The anti-PIP$_2$ immunogold labels inside of each area were counted and the labeling density for each subdivision and for an additional dendritic membrane area corresponding to the analyzed spine was determined using ImageJ/FIJI. In case labels were positioned directly at drawn border or separating lines, they were assigned to the area covered by the gold particle to a larger extend. All data obtained, including high values and zero profiles, were considered. No "outlier" analyses were conducted and no "outliers" were removed from the data collections, as the biological variance is reflected by the full range of determined labeling densities.

For time-resolved analyses of PIP$_2$ signals in spines and inhibitor studies, all results were normalized to the average labeling density detected in the overall spine of the steady-state condition (0 min).

Cluster analyses were conducted by using circular ROIs of a diameter of 100 nm. According to procedures established previously[22-24,77] clusters were defined as $n \geq 3$ gold particles per ROI.

**Statistics and reproducibility**. Statistical analyses were conducted using Graph-Pad Prism. If, using the Shapiro-Wilk-Test, the data were normally distributed, a Student's $t$-test (comparisons of two conditions), or a one-way analysis of variance (one-way ANOVA) with a subsequent Tukey's multiple comparisons test (comparison of more than two conditions) was performed.

If the Shapiro–Wilk test suggested a non-normal distribution, a Mann–Whitney test (for 2 conditions) or a Kruskal–Wallis test (≥3 conditions) with a subsequent Dunn's multiple comparison test was employed.

For two-factor analyses, two-way ANOVA was conducted in combination with either subsequent Šídák's or Bonferroni's multiple comparison tests.

**Reporting summary**. Further information on research design is available in the Nature Portfolio Reporting Summary linked to this article.

## Data availability
The authors declare that all data supporting the findings of this study are available within the paper and its supplementary information files. For source data reporting all individual data points underlying the quantitative analyses please see Supplementary Data 1.

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

## Acknowledgements

We thank K. Gluth, A. Kreusch, S. Linde, and M. Roeder for technical support. This work was supported by the IZKF and by the DFG (RTG1715 SP19 and KE685/7-1 to M.M.K. and QU116/9-1 to B.Q.).

## Author contributions

S.A.H.-M. and E.S. designed and performed experiments and interpreted the data. S.A.H.-M. co-wrote parts of the manuscript. M.W. provided technical advice and access to electron microscopy techniques and equipment. S.A.H.-M., E.S., and M.M.K. visualized data. M.M.K. conducted confirmatory, independent examinations demanded. B.Q. and M.M.K. conceived the project, designed experiments, interpreted the data, and wrote the manuscript.

## Funding

## Competing interests

The authors declare no competing interests.
