## [Peer Review File · Communications Biology]

Reviewers' comments:

Reviewer #1 (Remarks to the Author):

This manuscript examines PIP2 dynamics in LTD using a rapid cryofixation and immunolabeling approach. This approach has the advantage of avoiding probes that may impact the PIP2 dynamics themselves. I think this is a valuable contribution to the field. However, I do believe there are a few things that could be improved:

1. I would like to see more details on how the regions were defined on the EM images. The spines often appear somewhat unclear in the images, and I could easily see the defined border shifting depending on who is analyzing it. In many images there are particles sitting near the borders, so if it was shifted one way or the other it could easily influence the results. How did the authors adjust for errors in how the analysis regions were defined? Perhaps the authors could adjust for this by having multiple people analyze the data separately and compare the particle numbers determined by each person.
2. I would be curious to hear more about why the immunogold labeling of AMPA receptors and endocytosis markers was unsuccessful. This statement is not really explained in the text.
3. Initial percentage values are shown with the variation (e.g. line 166 "180±20%"), but later percentages do not have the variation shown (see lines 177, 195, 196, and others).
4. The authors briefly discuss the discrepancy between PIP2 levels measured with PLC-PH fluorescent probes vs. the immunogold method used in this study. I believe the authors should expand on this, as they rather quickly dismiss this discrepancy. The limitations that the authors directly mention are displacement with IP3, competing with PLC, and quenching PIP2 signals. It seems that these limitations would most strongly impact the measurement during stage 3, when the authors propose that PLC is active and therefore IP3 would be increasing. However, the discrepancy is observed 1 minute after NMDA simulation. Are there other factors that could cause this discrepancy? Perhaps due to competition with other PIP2 binding partners as the PIP2 is recruited into clusters and potentially involved in endocytic vesicle formation?
5. There are a number of issues with the English style that reduce the readability of the manuscript:
Line 66: should read "directly visualize PIP2"
Line 99: the second "can be" here is unnecessary
Line 144: I believe this should be "neighboring" instead of "neighbored"
Line 165: "maximum of PIP2" should be "maximum PIP2"
Line 178: "3times" needs a space: "3 times"
Line 299: "und inositol-1,4,5-triphosphat" should be "and inositol-1,4,5-triphosphate"
Line 337 is a bit unclear. It appears to be stating that the PIP2 content of the spine head is 40% higher than the general dendrite, but the way it is currently written it is unclear what the percentage refers to.
Line 341: should read "widely used as a PIP2 probe"
Line 344: I believe the word merely is unnecessary here
Line 393: "Additionally, also PIP2" can be shortened to "Additionally, PIP2"
Line 440: adding a comma to the number here would improve visibility (i.e., 200,000 instead of 200000
Line 508: P-face and E-face should be defined earlier in the text on first use

Reviewer #2 (Remarks to the Author):

The manuscript entitled "Long-term depression involves temporal and ultra-structural dynamics of phosphatidylinositol-4,5-bisphosphate relying on PIP5K, PTEN and PLC" written by Hofbrucker-MacKenzie et al. investigate the localization of PIP2 over the plasma membrane of cultured neurons by coupling rapid cryofixation and immunolabeling of freeze-fracture replicas.

This study aims to obtain evidence of PIP2 involvement as signaling cues for synaptic plasticity in the postsynaptic compartments. For this aim, authors quantified PIP2 immunogold labeling across subdomains of the neuronal plasma membrane after induction of long-term synaptic depression (LTD), one form of synaptic plasticity. They also investigated the regulatory process of the PIP2 level changes after the LTD induction. The overall design of the study is well planned and conducted. They succeeded in detecting a rapid and transient increase of the PIP2 labeling at the specific subareas of dendritic spines after induction of LTD and in identifying enzymes responsible for the rise and decay. On the other hand, there are several concerns about the specificity and sensitivity of labeling for PIP2 in their detection system, which make interpreting the presented results difficult and their conclusions not fully sound. Thus, I think this manuscript needs significant revision before acceptance for publication in Communications Biology.

Major concerns:

1. Labeling specificity of the used antibody to PIP2 is not fully established.

Authors tested the labeling specificity to PIP2 by using liposomes containing PIP2 or PS, but not with other structurally similar phosphoinositides such as PI4P and PI3,4,5P. In biological membranes, these similar phosphoinositides co-exist, and the antibody may produce labeling by its cross-reactivity to these related molecules. Thus, authors should confirm no antibody reactivity to other phosphoinositide species so that labeling results can be interpreted as an index of PIP2 expression.

2. Labeling sensitivity in this study seems to be very low.

Although authors demonstrated that liposomes with PIP2 showed a more than 16-fold higher immunogold labeling density than those with PS, its labeling density is Ca. <3 labeling / μm^2 . Based on a report on the molecular arrangement of liposomes by Huang and Mason (PNAS, 1978) and an expected head area of PIP2 as 0.7 nm^2 , phosphatidylcholine vesicles with 5%PIP likely contain PIP2 at about 70000 molecules / μm^2 . Given that this density is true, the labeling sensitivity of the authors' method would be estimated as less than 0.004% and PIP2 distribution over the plasma membrane, in reality, would be totally different from what we see in the images provided in Figures and thus conclusions drawn in the manuscript would become different from current ones. Thus, I think that the evaluation of detection sensitivity to PIP2 by the authors method should be conducted and conclusions from individual experiments should be drawn by taking it into account.

3. In relation to the above concern, this reviewer also wonders why authors used immunogold particles with 10 nm or 15 nm colloidal gold. Such large immunogold often resulted in lower detection sensitivity than those with 5 nm particles. Thus, authors should use the smaller immunogold for higher labeling efficiency to visualize more reliable PIP2 distribution.

4. Biological significance of the PIP2 increase in the upper and middle parts of the dendritic spine head is unclear, and this would be solved by co-labeling for synaptic molecules.

Synaptic contacts could be established anywhere at the spine heads, and changes in PIP2 concentration would occur inside and vicinity of synaptic sites by NMDA receptor activation. Thus, it is more important to show the spatial relationship between PIP2 accumulation and synaptic sites by double labeling for PIP2 and PSD-95 or NMDARs.

Minor concerns:

1. It is not mentioned in the text if no aggregation of the secondary antibody (antibody clumping) was confirmed or not. If such examination was not conducted, please check if aggregation of the antibody is present or not and modify the manuscript accordingly.

2. The labeling densities for PIP2 shown in Figures 1i, 1m, 2d, and 2e are suggesting the occurrence of significant amounts of 0-labeling profiles, especially for spines and their subareas, which would hamper the proper assessment of the abundance of target molecules. However, all of the numerical and graphical data presented in the manuscript are shown as mean \pm SEM. Thus, readers including

me encounter difficulty or cannot assess the adequacy of data analysis in this manuscript. I recommend that authors use a box chart format for data presentation.

3. The acquisition protocol of sample images should be explained in detail.

I am curious how authors handled the 0-labeling profiles at the image acquisition. Did authors avoid imaging such profiles or include all of the profiles regardless of the labeling intensity? Please explain clearly about this in the main text.

Point-to-point responses to the Reviewers' comments

Reviewer #1 (Remarks to the Author):

This manuscript examines PIP₂ dynamics in LTD using a rapid cryofixation and immunolabeling approach. This approach has the advantage of avoiding probes that may impact the PIP₂ dynamics themselves. I think this is a valuable contribution to the field. However, I do believe there are a few things that could be improved:

1. I would like to see more details on how the regions were defined on the EM images. The spines often appear somewhat unclear in the images, and I could easily see the defined border shifting depending on who is analyzing it. In many images there are particles sitting near the borders, so if it was shifted one way or the other it could easily influence the results. How did the authors adjust for errors in how the analysis regions were defined?

Perhaps the authors could adjust for this by having multiple people analyze the data separately and compare the particle numbers determined by each person.

As stated in the manuscript, the concerns the reviewer brings up reflect general concerns that we had considered when we started our quantitative analyses.

i) Therefore, all changes of PIP₂ during LTD induction were studied in a fully blinded manner.

ii) Individual judgement differences concerning area definition (especially when labeling densities are evaluated in relatively small membrane areas) can also contribute to the noise in the quantitative data sets of labeling densities (although the reviewer will agree with us that actually the variance within sets of biological samples usually is a major source of noise in quantitative examinations in life sciences). We therefore defined a variety of standardized rules and parameters for these evaluations. The originally submitted manuscript did not extensively describe the procedure due to space limitations. **As requested by the reviewer, the revised manuscript now describes the procedure in detail (please see revised Material and Methods).**

The robustness of our findings are also highlighted by the fact that the more than doubling of the PIP₂ labeling densities in dendritic spines during LTD induction (0 vs. 3+7) were reproducibly seen in several independently obtained data sets.

As far as exact reproducibility by several experimenters is concerned, we have embarked on such an endeavor during the revision work of this study. The revised manuscript now includes a comparison of anti-PIP₂ labeling densities of control (0 min) vs. LTD induction (3 min NMDA+7 min) determined by 2 independent experimenters on about 60 images for each condition.

The data set reflects key findings presented in Figure 3 and Figure 4 and is presented in absolute numbers of labeling densities for detailed comparability. **Importantly, the mean labeling densities independently determined by two experimenters for i) total spines, ii) spine heads, iii) base, iv) neck, v) lower head, vi) middle head and vii) upper head were virtually identical (newly added Supplementary Figure S3).**

This consistency between the data from the two different experimenters is remarkable, as the second analysis was done completely independently and the second experimenter furthermore did not receive any kind of training by the main author but solely the instructions reported in the **revised Material and Methods**.

In detail, our further individual data point comparisons for i) gold labels counted, ii) assignments of labels to subareas, iii) (sub)area definition and iv) sub)area measurement between all (58+59)x7=819 areas (experimenter 1) and (59+60)x7=833 areas (experimenter 2), respectively, defined and

analyzed revealed that independent evaluations by different experimenters led to virtually the same data for spine selection, spine area determinations and labeling counts and area assignment of labeling (our unpublished individual data point analysis).

The minimal deviations between the new experimenter 2 data and the trained experimenter 1 data that we were able to identify mostly reflected minor differences in lateral base border definitions as well as few-nm differences in the positioning of the head borders but these minor uncertainties were apparently averaged out well. Furthermore, we noted one difference at the level of spine selection: in each of the two conditions, the new experimenter 2 identified 1 additional spine to be analyzed in the 58 control images and 59 3+7 min images (our unpublished individual data point analysis).

Taken together, our extensive quantitative EM efforts conducted as additional revision work demonstrate in a very impressive manner that with receiving solely the instructions for the standardized evaluation procedure reported in the revised Material and Method section and no further training, even a new experimenter can conduct quantitative examinations leading to the same insights in PIP₂ signals in LTD inductions as those obtained by the original, trained experimenter.

2. I would be curious to hear more about why the immunogold labeling of AMPA receptors and endocytosis markers was unsuccessful. This statement is not really explained in the text.

The revised manuscript now contains more extensive information on this. As already reported in the originally submitted manuscript, it was not possible to establish double-immunogold labeling procedures of anti-PIP₂ labeling at the P-face of neuronal membranes with that of AMPA receptors or endocytic components, such as clathrin and dynamin.

Extensive, further experiments conducted during our revision work suggest that also immunogold labeling of NMDA receptors and of postsynaptic scaffold proteins, such as PSD95 and ProSAP/Shanks, are difficult or impossible to establish at freeze-fractured membranes of dendritic spines. **These efforts are briefly described and discussed in the Results section of the revised manuscript (prior to Fig. 5 results description).**

As alluded to in the revised manuscript, the difficulties seem to be manifold: In line with our own difficulties to label NMDA and AMPA receptors at the P-face together with PIP₂, both NMDA and AMPA receptors were reported in several publications to be detected at the extracellular side after membrane facturing (Rash et al 2005; Antal et al 2008; Tarusawa et al 2009; Rubio et al. 2014; Rollenhagen et al, 2022). **This additional literature is now cited as new Ref.#33-37 in the revised manuscript.**

The difficulties to detect proteins, which are merely indirectly or merely peripherally associated to membranes but not inserted (clathrin, dynamin, PSD-95, ProSAP/Shanks and others), are probably due to the sample preparation procedure, which provides full membrane access by effectively removing such merely peripherally associated material - a prerequisite for reliable detection of lipid cues in the membrane. **Also these aspects are now more elaborated on in the revised manuscript.**

We hope the reviewer is content with the summary on our efforts and with the additional literature added.

3. Initial percentage values are shown with the variation (e.g. line 166 “180±20%”), but later percentages do not have the variation shown (see lines 177, 195, 196, and others).

Thank you. **This inconsistency in writing has been fixed in the revised manuscript.**

4. The authors briefly discuss the discrepancy between PIP₂ levels measured with PLC-PH fluorescent probes vs. the immunogold method used in this study. I believe the authors should expand on this, as they rather quickly dismiss this discrepancy. The limitations that the authors directly mention are displacement with IP₃, competing with PLC, and quenching PIP₂ signals. It seems that these limitations would most strongly impact the measurement during stage 3, when the authors propose that PLC is active and therefore IP₃ would be increasing. However, the discrepancy is observed 1 minute after NMDA simulation. Are there other factors that could cause this discrepancy? Perhaps due to competition with other PIP₂ binding partners as the PIP₂ is recruited into clusters and potentially involved in endocytic vesicle formation?

We acknowledge that the previously submitted manuscript may have been a bit too short in the discussion of the limitations of expressing PLC-PH as a probe for PIP₂ instead of cryopreserving the PIP₂ distribution and then detecting PIP₂ on freeze-fractured membranes using specific immunogold-conjugated antibodies, as in the method we established for the ultra-high resolution visualization of lipid cues.

The reviewer is right, the limitations of expressing PLC-PH as a probe for PIP₂ do not only include the previously mentioned ones but also include competitions with other PIP₂ binding partners, which will thereby be decoupled from PIP₂ signals. This is what we meant by “quenching PIP₂ signals” in the previous manuscript. **We hope the reviewer will be content with the extended description and more elaborate explanations provided in the revised manuscript.** It now reads:

“Importantly, exogenous expression of reporter proteins, such as PLC-PH, although being widely used as PIP₂ probe, furthermore does not unambiguously report PIP₂ level changes because PLC-PH has been shown to be displaced from PIP₂ by the increase of PIP₂’s intracellular degradation product IP₃^{6,46}. This limitation is severe, as the half-life of PIP₂ in cells merely is only about 1 min^{47,48}. PLC-PH expression furthermore impacts cell physiology and signaling by competing with the PIP₂-degrading enzyme PLC and by quenching PIP₂ signals^{49,50}, as PLC-PH bound to PIP₂ will compete with the binding of PIP₂ effector proteins, such as endocytic, cytoskeletal and signaling components⁷ and thereby hamper proper recognition and transmission of PIP₂ signals.”

5. There are a number of issues with the English style that reduce the readability of the manuscript:

Line 66: should read “directly visualize PIP₂”

Thank you for noticing this typo. **The typo was corrected in the revised manuscript.**

Line 99: the second “can be” here is unnecessary

Thank you for noticing this error. **The sentence has been changed in the revised manuscript.**

Line 144: I believe this should be “neighboring” instead of “neighbored”

Thank you. There seems to be no adjective for this verb in English but most appropriate seems to be “adjacent”. **We have corrected the revised manuscript accordingly.**

Line 165: “maximum of PIP2” should be “maximum PIP2”
Thank you. **The sentence has been shortened in the revised manuscript.**

Line 178: “3times” needs a space: “3 times”
Thank you. **The missing space has been added in the revised manuscript.**

Line 299: “und inositol-1,4,5-triphosphat” should be “and inositol-1,4,5-triphosphate”
Thank you. **This typo was corrected in the revised manuscript.**

Line 337 is a bit unclear. It appears to be stating that the PIP2 content of the spine head is 40% higher than the general dendrite, but the way it is currently written it is unclear what the percentage refers to.
Thank you. **The missing “plus” has been added in the revised manuscript.**

Line 341: should read “widely used as a PIP2 probe”
Thank you for noticing this. **The missing verb has been added in the revised manuscript.**

Line 344: I believe the word merely is unnecessary here
Thank you. **The sentence has been changed in the revised manuscript.**

Line 393: “Additionally, also PIP2” can be shortened to “Additionally, PIP2”
Thank you. **The sentence has been shortened accordingly in the revised manuscript.**

Line 440: adding a comma to the number here would improve visibility (i.e, 200,000 instead of 200000
Thank you for this suggestion. This is certainly true for Americans, however, for many other people world-wide, the comma is read as a comma (point) and the number could therefore be confused with 200,0, i.e. 200.0. Also, we figured that it is important to keep the consistence of writing with other numbers in the manuscript (which also do not contain any separation of digitals representing 1000s and those for 100s, 10s and single digits).

Line 508: P-face and E-face should be defined earlier in the text on first use
Thank you. **P- and E-face are now explained upon first mention in the revised manuscript.**

Reviewer #2 (Remarks to the Author):

The manuscript entitled “Long-term depression involves temporal and ultra-structural dynamics of phosphatidylinositol-4,5-bisphosphate relying on PIP5K, PTEN and PLC” written by Hofbrucker-MacKenzie et al. investigate the localization of PIP₂ over the plasma membrane of cultured neurons by coupling rapid cryofixation and immunolabeling of freeze-fracture replicas.

This study aims to obtain evidence of PIP₂ involvement as signaling cues for synaptic plasticity in the postsynaptic compartments. For this aim, authors quantified PIP₂ immunogold labeling across subdomains of the neuronal plasma membrane after induction of long-term synaptic depression (LTD), one form of synaptic plasticity. They also investigated the regulatory process of the PIP₂ level changes after the LTD induction. The overall design of the study is well planned and conducted. They succeeded in detecting a rapid and transient increase of the PIP₂ labeling at the specific subareas of dendritic spines after induction of LTD and in identifying enzymes responsible for the rise and decay. On the other hand, there are several concerns about the specificity and sensitivity of labeling for PIP₂ in their detection system, which make interpreting the presented results difficult and their conclusions not fully sound. Thus, I think this manuscript needs significant revision before acceptance for publication in Communications Biology.

Major concerns:

1. Labeling specificity of the used antibody to PIP₂ is not fully established.

Authors tested the labeling specificity to PIP₂ by using liposomes containing PIP₂ or PS, but not with other structurally similar phosphoinositides such as PI4P and PI3,4,5P. In biological membranes, these similar phosphoinositides co-exist, and the antibody may produce labeling by its cross-reactivity to these related molecules. Thus, authors should confirm no antibody reactivity to other phosphoinositide species so that labeling results can be interpreted as an index of PIP₂ expression.

In the revised manuscript, we have now replaced our antibody characterization analyses with liposomes (former Figure 1a-c) by new, more extended experiments (revised Figure 1a-e). The new experiments done in parallel to ensure full comparability of all conditions testes demonstrate the specificity of labeling, as the replaced ones before, and in addition now also address putative crossreactions to other phosphoinositides, in particular to PI(3,4,5)P₂, as requested by the reviewer. PIP₃ also has been implicated in synaptic remodeling processes. It may therefore indeed be necessary to carefully distinguish signaling cues provided by PIP₂ from those provided by PIP₃ **(revised Figure 1a-e).**

The new experiments clearly demonstrate that the anti-PIP₂ immunodetection in lipid bilayers is specific for PI(4,5)P₂ **(revised Fig. 1a-e)**. Quantitative electron microscopic analyses furthermore show that neither PI(3,4,5)P₃ nor PI(3,4)P₂ were recognized by the anti-PIP₂ antibodies from Enzo Life Sciences **(revised Fig. 1e)**. **Thus, even phosphoinositides that contain all antigenic features also present in PIP₂ but either offer an additional feature (as in PIP₃) or display one of these in a different position (as in PI(3,4)P₂) are recognized by the anti-PIP₂ antibodies. Instead, the established anti-PIP₂ immunodetection clearly was specific for PI(4,5)P₂ (PIP₂).**

In addition, we would like to refer the reviewer to the fact that, besides the characterization with liposomes of defined composition **(revised Figure 1a-e)**, we have also extensively characterized the specificity of the anti-PIP₂ immunogold labeling by further lines of experimentation using cells. These efforts clearly distinguished the anti-PIP₂ labeling from unspecific background labeling by quantitative assessment of labeling of ice control surfaces and of E- instead of P-faces of the plasma membranes. Importantly, the neuronal experimentation even included a quantitative analysis of an antibody quenching with the antigen PIP₂ – a powerful control rarely attempted and shown in other published characterizations of commercially available antibodies.

2. Labeling sensitivity in this study seems to be very low.

Although authors demonstrated that liposomes with PIP₂ showed a more than 16-fold higher immunogold labeling density than those with PS, its labeling density is Ca. <3 labeling / μm^2 . Based on a report on the molecular arrangement of liposomes by Huang and Mason (PNAS, 1978) and an expected head area of PIP₂ as 0.7 nm², phosphatidylcholine vesicles with 5%PIP likely contain PIP₂ at about 70000 molecules / μm^2 . Given that this density is true, the labeling sensitivity of the authors' method would be estimated as less than 0.004% and PIP₂ distribution over the plasma membrane, in reality, would be totally different from what we see in the images provided in Figures and thus conclusions drawn in the manuscript would become different from current ones. Thus, I think that the evaluation of detection sensitivity to PIP₂ by the authors method should be conducted and conclusions from individual experiments should be drawn by taking it into account.

The theoretical calculation of the reviewer is of course right, with a lipid head area occupation of 0.7 nm², one detection by antibody would be able to sterically cover a membrane area that could theoretically easily represent many PIP₂ molecules if one would assume a maximally dense PIP₂ packaging and if one furthermore takes into account the hydrodynamic radius of this or other protein-based probes. However, our study of course did not aim at determining the exact localization of each and every PIP₂ molecule in a given neuron. This would have been absolutely unrealistic and we therefore did not intend to make any claims in the direction of determining the absolute levels of PIP₂ signals in neuronal membrane compartments and subdomains. Wherever this may not have been clear in the previous manuscript, we hope that the reviewer will agree with us that **the revised manuscript is now very clear and precise on this, as it at several places specifies that relative changes of PIP₂ signals are quantified and tracked at different membrane areas at ultra-high resolution.**

As far as specifically the liposome experiments previously shown in Figure 1c, which the reviewer referred to, are concerned, these data sets have been replaced by a more extended examinations of anti-PIP₂ immunolabeling specificity (revised Figure 1a-e; see above, point 1 of the reviewer). This additional experimentation addressing putative crossreactivities of the anti-PIP₂ antibodies shows an anti-PI(4,5)P₂ labeling density that is about an order of magnitude higher than the one of the replaced experiment the reviewer criticized (**revised Figure 1e**).

Our results indicate that actually parameters other than the geometric ones used for the reviewer's calculation are of high importance in *in vitro*-experimentations. We would like to suggest that one of these aspects is the hydrolysis of PIP₂ during the lengthy process of liposome generation – especially since this process has to be conducted at 37°C. In line, reducing the time of liposome formation, as done for the new experimentation, led to increased anti-PIP₂ labeling density at the liposome membranes.

Importantly, this caveat in *in vitro* experimentations with PIP₂ is circumvented by the quick-freezing protocol used for all experimentation with primary neurons. With a cooling rate of about 4000 K/s, even short-lived lipid signals will be fully preserved and importantly also shown at unchanged positions in the set of membrane subareas we analyzed in control neurons and in neurons subjected to LTD induction.

3. In relation to the above concern, this reviewer also wonders why authors used immunogold particles with 10 nm or 15 nm colloidal gold. Such large immunogold often resulted in lower detection sensitivity than those with 5 nm particles. Thus, authors should use the smaller immunogold for higher labeling efficiency to visualize more reliable PIP₂ distribution.

Strong dependences of classical immunogold labelings of thin sections are indeed a serious issue in quantitative TEM studies, as an increase of probe sizes by only 5 nm can in extreme cases mean that one order of magnitude lower labeling densities are obtained or that the immunogold labeling may even be lost completely. We of course also were concerned that the labeling of freeze-fractured samples may be hampered by the same effects, as we figured that immunolabelings should in our cases better not be done with very small probes, such as 5 nm gold conjugates, for two main reasons: i) the very high magnifications needed to reliably detect 5 nm gold would preclude the examination of larger membrane areas, such as those of a complete dendritic spine and neighboring dendritic areas. The systematic exploration of large membrane surfaces, however, is a prerequisite for doing a reliable quantitative distribution study and actually is one of the advantages of freeze-fracturing procedures, as freeze-fracturing yields such large membrane surfaces to evaluate.

ii) 5 nm gold can more easily be overlooked, if located in the more electron-dense areas created by platinum shadowing of either membrane topologies or of membrane proteins protruding from the membrane surface. Detection uncertainties would create another source of potential error and further variance in quantitative examinations of labeling densities.

As the use of larger immunogold avoids such difficulties, **we have conducted comparative, parallel anti-PIP₂ immunogold labeling experiments with 5 nm, 10 nm and 15 nm colloidal gold-conjugated secondary antibody. The quantitative examinations showed that there were no statistically significant differences in the labeling densities detected irrespective of which gold size was used (see newly added Supplementary Figure S2).**

These results probably reflect the undisturbed accessibility of freeze-fractured membrane surfaces for probes – a fundamental difference when compared to e.g. classical resin-embedded EM samples subjected to immunogold labeling. In line with our results, also beautiful work of the Fujimoto lab (Fujita et al. 2009 *PNAS*; former Ref.#36, now changed to Ref.#21 in the revised manuscript) suggested that steric hindrances of protein-based probes (antibodies in our study; recombinant GST-fusion proteins in Fujita et al. 2009) are generally low at freeze-fractured membranes.

4. Biological significance of the PIP₂ increase in the upper and middle parts of the dendritic spine head is unclear, and this would be solved by co-labeling for synaptic molecules.

Synaptic contacts could be established anywhere at the spine heads, and changes in PIP₂ concentration would occur inside and vicinity of synaptic sites by NMDA receptor activation. Thus, it is more important to show the spatial relationship between PIP₂ accumulation and synaptic sites by double labeling for PIP₂ and PSD-95 or NMDARs.

Our quantitative electron microscopic examinations indeed show that the most pronounced transient increase in PIP₂ labeling density occurs in the middle and upper spine head areas (Figure 4). The doubling of PIP₂ in the membrane areas of the lower spine head, in contrast, remained statistically insignificant (Figure 4). This preference for the upper 2/3 of the spine head membrane most likely reflects the simple fact that usually these more distal areas of dendritic spines are contacted by presynapses. As already reported in the originally submitted manuscript, it was not possible to establish double-immunogold labeling procedures for PIP₂ together with AMPA receptors.

Extensive, further experiments conducted during our revision work showed that also immunogold labeling of NMDA receptors at postsynaptic P-faces is difficult to obtain. We consistently observed this difficulty with several commercially available antibodies. Our observations hereby are in line with the fact that both NMDA and AMPA receptors were detected at the extracellular side after membrane fracturing of excitatory glutamatergic postsynapses (Rash et al 2005; Antal et al 2008; Tarusawa et al 2009; Rubio et al. 2014; Rollenhagen et al, 2022). This additional literature is now cited as new Ref.#33-37 in the revised manuscript.

The difficulties to detect proteins, which are merely indirectly or merely peripherally associated to membranes but not inserted (such as the scaffold protein PSD-95 suggested by the reviewer; but according to our observations, also ProSAP/Shanks and others), **seems to be due to the sample preparation procedure, which provides full membrane access by effectively removing such merely peripherally associated material - a prerequisite for reliable detection of lipid cues in synaptic membrane areas.**

Both of these aspects are now more elaborated on in the revised manuscript. We hope the reviewer is content with the summary on our efforts and with the additional literature added.

Minor concerns:

1. It is not mentioned in the text if no aggregation of the secondary antibody (antibody clumping) was confirmed or not. If such examination was not conducted, please check if aggregation of the antibody is present or not and modify the manuscript accordingly.

Putative antibody aggregations can indeed represent an issue in the establishment of immunogold labelings. We have used different secondary antibodies from BBI solutions before and did not have any difficulties with antibody aggregations under our conditions (e.g. Seemann et al., 2017 eLIFE; Wolf et al., 2019 Nat. Cell Biol.; Izadi et al., 2021 J. Cell Biol.).

Antibody aggregations would also readily be visible in standard secondary antibody control incubations. They usually appear as very large, dense clouds of many gold labels with further associated electron-dense material. The specific anti-PIP₂ clusters we detected were marked by only few gold particles (Figure 5). Antibody aggregates would furthermore not be restricted to P-face membrane areas but would occur at all types of surfaces in the specimen.

Thus, even if one or two of such aggregations would have been found at some places and by coincidence even exactly at a dendritic spine membrane, they would probably have been easily recognizable and could have been excluded from analysis. However, as stated above, we did not observe any difficulties with antibody aggregates.

2. The labeling densities for PIP₂ shown in Figures 1i, 1m, 2d, and 2e are suggesting the occurrence of significant amounts of 0-labeling profiles, especially for spines and their subareas, which would hamper the proper assessment of the abundance of target molecules. However, all of the numerical and graphical data presented in the manuscript are shown as mean +/- SEM. Thus, readers including me encounter difficulty or cannot assess the adequacy of data analysis in this manuscript. I recommend that authors use a box chart format for data presentation.

The section Statistical Analysis in the Material and Method section of the manuscript explicitly stated that **all quantitative data of the study have of course been analyzed for normal data distribution using the Shapiro-Wilk-Test prior to selection of the appropriate test for statistical analysis.** Thus, the adequacy of our data analysis is ensured.

Zero-labeling profiles were of course not excluded from the analysis. Please also see our answer below for further explanations and changes in the revised manuscript concerning this point (Point Minor 3. of the reviewer).

We would also like to note that all numerical data is available in the **Supplementary Table 1 data compilation, which is part of the uploaded revised manuscript** and will be published along with all other information.

As far as the presentation of data is concerned, we acknowledge that in studies with very few data points, it is helpful to see the individual data points directly in the figure and not in a table. Yet, our data is not hampered by low n numbers. Relatively low n numbers can be found in the in vitro experimentation with liposomes. **We have therefore added a bar/dot plot presentation of this data set to the revised manuscript (newly added Supplementary Figure S1).**

In general, we would like to point out that showing the full range of data distribution in EM labeling density analyses instead of focusing at the means is not always useful for the reader, as neither the amount of zero profiles can be accurately represented at the ground line nor differences between different conditions may well be visualizable. Yet, if deemed necessary, we shall be happy to discuss including more figure panels visualizing the distribution of all individual data points listed in the numerical data compilation.

3. The acquisition protocol of sample images should be explained in detail. I am curious how authors handled the 0-labeling profiles at the image acquisition. Did authors avoid imaging such profiles or include all of the profiles regardless of the labeling intensity? Please explain clearly about this in the main text.

We would like to refer the reviewer to the Material and Method section of the manuscript. It clearly stated that analyses were done in a blinded manner, that images were obtained by systematic screening of the grids and that only spines shorter than 0.75 μm or longer than 2 μm as well as images lacking the dendritic attachment of the spines were excluded from the analysis. Thus, all morphological structures that could clearly be categorized as mushroom-type spines (as well as neighboring dendrite segments) were used for the analysis.

As the reviewer requested, the revised manuscript now states in both Material and Methods and in the main text that all mushroom-type spines were analyzed.

In quantitative immunogold labeling density determinations, zero profiles of labeling usually do not only occur in control samples but can also occur in other conditions evaluated - especially if examined (sub)areas are small. Zero profiles thus are part of the expectable data distribution. This also applies to all membrane areas examined in our study – may it be liposomes, cell soma membrane areas as well as dendrites or mushroom spines of neurons (see numerical data compilation for all quantitative figure panels in Supplementary Table 1). In order to determine anti-PIP₂ labeling densities at different membrane areas, it would be scientifically inappropriate to consider the observed anti-PIP₂ labeling density at a certain location as parameter in spine sampling and/or to exclude zero profiles.

The revised manuscript now explicitly states that all data obtained, including high values and zero profiles, were considered and that no “outlier” analyses were conducted and no “outliers” were removed from the data collections, as the biological variance is reflected by the full range of determined labeling densities.

Zero profile inclusion now also is explicitly mentioned in the section of anti-PIP₂ immunogold labeling establishment (please see the Material and Method section of the revised manuscript).

REVIEWERS' COMMENTS:

Reviewer #1 (Remarks to the Author):

The authors have addressed my concerns, and I believe this manuscript will make a valuable contribution to the field.

Reviewer #2 (Remarks to the Author):

I thank the authors for their thoughtful responses to my review comments. My concerns about the specificity and sensitivity of PIP2 labeling and the detail of the quantification procedures and statistical analysis of labeling, which I raised in the previous review, have been significantly improved by the authors' additional experiments and revisions to the manuscript, making the current manuscript much easier to understand and supporting the authors' conclusions. Therefore, I now think this manuscript is suitable for Communications Biology.